# CASE: Challenger Arm Sampling for Efficient In-Context Reasoning

## Abstract

The in-context learning paradigm with LLMs has been instrumental in advancing applications that require complex reasoning over natural language. An optimal selection of few-shot examples (exemplars) is essential for constructing effective prompts under a limited budget. In this paper, we frame the problem of exemplar selection for In-Context Reasoning (ICR) as a top-$m$ best arms identification problem. A key challenge in this context is the exponentially large number of arms that need to be evaluated to identify the $m$-best arms. We propose CASE (**C**hallenger **A**rm **S**ampling for **E**xemplar selection), a novel *selective exploration* strategy that maintains a shortlist of "challenger" arms, which are current candidates for the top-$m$ arms. In each iteration, only the arms from this shortlist and the current top-$m$ set are pulled, thereby reducing sample complexity and, consequently, the number of LLM evaluations. Furthermore, we model the scores of exemplar subsets (arms) using a parameterized linear scoring function, leading to a *stochastic linear bandits* setting. In this setting, CASE identifies the top-$m$ arms with significantly fewer evaluations than existing state-of-the-art methods. CASE effectively works with black box LLMs and selects a static set of few-shot examples, resulting in an extremely efficient scheme for in-context reasoning. The exemplars selected with CASE show surprising performance gains of up to **15.19%** compared to state-of-the-art exemplar selection methods. We release our **code and data** [1].

## 1 Introduction

In-context learning (ICL) and Chain-of-Thought (COT) have emerged as important techniques for enhancing the capabilities of large language models (LLMs) across a range of tasks like question answering and complex reasoning. ICL allows LLMs to perform tasks by conditioning on a context that includes examples or instructions, without the need for additional fine-tuning, making it flexible and adaptable. COT facilitates complex reasoning by decomposing tasks into intermediate steps, and employing rationales explaining the reasoning process to the LLM. However, one key challenge in maximizing the effectiveness of ICL is the careful selection of few-shot examples along with their corresponding rationales (Lu et al., 2022; Zhao et al., 2021).

The inclusion of rationales alongside standard training examples has been found to be especially important for complex reasoning tasks, such as multistep reasoning-based QA (Geva et al., 2021b; Chen et al., 2022b; Lu et al., 2023b). We refer to the triplet of *(input, rationale, output)* as an exemplar and define the paradigm of using rationale-augmented examples in ICL as *In-Context Reasoning (ICR)*. The majority of existing approaches for exemplar selection rely on heuristics or trial-and-error methods Fu et al. (2023); Brown et al. (2020a), with only a few attempting to address the problem in a more principled way Xiong et al. (2024). Exemplars selected for ICR can be classified into two categories: *task-level*, where a static set of exemplars representative of the task is chosen for inference, and *instance-level*, where exemplars are dynamically selected for each test instance during inference. Instance-level selection approaches typically use similarity (Rubin et al., 2022; Xiong et al., 2024) and diversity-based measures (Ye et al., 2023b) to identify the most suitable exemplars from the training pool for each test instance, which introduces overhead during inference. Additionally, these approaches do not consider the positive and negative interactions between the exemplars, as they select each exemplar independently. In contrast, selecting a static set of exemplars at the task level not

---

[1] https://anonymous.4open.science/r/CASE_exemplar_bandits-7403

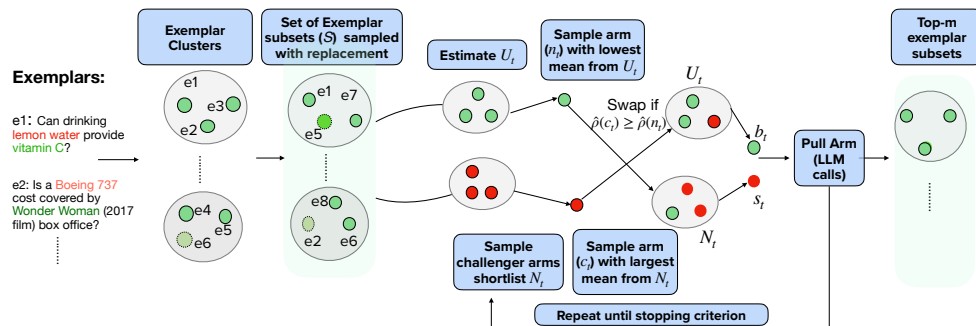

Figure 1: **Overview of CASE for selection of top-$m$ best exemplar subsets (arms)**. The process begins with clustering exemplars from the training set, followed by sampling with replacement to form exemplar subsets ($\mathcal{S}$). In each iteration, the top-$m$ arms ($U_t$) and challenger arms ($N_t$) are computed from a uniformly sampled set. Each arm pull corresponds to an LLM evaluation using the exemplar subset. The process repeats until the stopping criterion is met.

only eliminates inference-time overhead but also enables prompt caching, allowing for the reuse of key-value (KV) attention states Gim et al. (2024). In this work, we propose a principled approach for task-level exemplar selection. The closest work that proposes a task-level exemplar approach is LENS Li & Qiu (2023), which incurs high computational costs by valuing each exemplar in the training pool using an informativeness measure that requires LLM call over multiple iterations. Additionally, LENS relies on confidence measures from the LLM, making it difficult to extend to black-box models, and does not account for interactions between exemplars. In contrast, we implicitly capture interactions by scoring exemplar subsets as a whole rather than evaluating them independently.

One of the challenges in proposing a principled solution is the limited understanding of the underlying mechanisms of in-context learning. Recent studies aimed at developing theoretically grounded models for in-context learning (Zhang et al., 2023a) primarily focus on linear functions and linear transformers. Therefore, to achieve a principled and efficient selection of exemplars, we require a surrogate model for modeling the goodness of an in-context learning procedure. In this paper, we use a linear function based on sentence similarities between the in-context examples and validation examples as the surrogate model for the goodness score. This approach also be interpreted as a reward model for a multi-armed bandit-based exemplar selection scheme, leading to the linear stochastic bandits setting (Abbasi-Yadkori et al., 2011).

We formulate the selection of top-$m$ exemplar subsets as the problem of identification of top-$m$ arms (Réda et al., 2021) in the stochastic linear bandit setting (Réda et al., 2021). The GIFA framework (Réda et al., 2021) proposed an efficient gap-index based top-$m$ arms identification algorithm with reduced sample complexity. However, the computation of *challenger arm* (current candidates for top-$m$ arms), requires computation of gap-indices between all currently estimated top-$m$ arms and the remaining arms. This is impractical in our setting since each arm corresponds to a $k$-sized subset of the training exemplar set, leading to an exponential number of candidate arms. Hence we need an algorithm that can sample arms from the candidate sets. While some uniform sampling algorithms for top-$m$ arms identification exist, Chen et al. (2017); Kaufmann & Kalyanakrishnan (2013) for the general multi-arm bandit setting, to the best of our knowledge, there are no sampling-based algorithms for identification of top-$m$ arms in the stochastic linear bandits setting.

In this work, we propose CASE (**C**hallenger **A**rm **S**ampling for **E**xemplar Selection) where we propose a principled sampling of challenger arms to form a shortlist challenger set, pruning the space of possible candidate arms (see fig. 1). Our key idea is to iteratively create a low-regret set of selected challenger arms, in addition to the current top-$m$ arms, from uniformly sampled arms. This leads to a selective exploration-based algorithm. We concurrently apply the state-of-the-art gap-index-based algorithm rule for selecting top-$m$ arms out of the total exploration set. We also provide theoretical arguments to justify our novel approach of combined selective exploration and gap-index based identification of top-$m$ arms. When applied to exemplar selection, we observe improvements in task performance of upto **15.19%** and reduces number of LLM calls (about **10.5x**) when compared to state-of-the-art exemplar selection approaches. To summarize, our key contributions are:

- We propose a novel and principled gap index, MAB based approach for task level exemplar selection. Our contribution primarily lies in principled challenger arm sampling to prune the search space, which renders the process highly efficient compared to state-of-the-art MAB frameworks and other subset, exemplar selection approaches.

- We demonstrate that CASE has significantly lower number of comparisons, gap-index computations and runtime compared to state-of-the-art gap-index based algorithms.

- We perform extensive experiments on diverse tasks that require complex reasoning and observe that CASE is sample efficient and yields high gains compared to other state-of-the-art exemplar selection approaches.

## 2 RELATED WORK

**Exemplar Selection for ICL.** The rise of LLMs has transformed them into general-purpose answering engines through emergent capabilities like ICL (Brown et al., 2020b; Wei et al., 2022; 2023; Wang et al., 2023a; Kojima et al., 2023; Chen et al., 2022a) where a few examples are provided to LLMs to demonstrate the task. To eliminate manual selection, several automated methods have emerged, such as reinforcement learning (Zhang et al., 2022; Lu et al., 2023a), trained retrievers Xiong et al. (2024), Determinantal Point Processes (Ye et al., 2023a) and constrained optimization (Tonglet et al., 2023), which are effective for reasoning tasks. Additionally, dynamic selection methods that are learning-free, such as similarity-based (Rubin et al., 2022), complexity-based (Fu et al., 2023), and MMR (Ye et al., 2023b), have been explored. However, dynamic selection methods increase inference-time computational costs. To address this, a pre-selected, representative set of exemplars can be chosen for ICL, akin to coreset selection methods (Guo et al., 2022), though the key difference is that ICL does not involve parameter updates, unlike traditional deep learning training. To the best of our knowledge, there has been very little research in a principled, efficient approach for task-level exemplar selection, with the closest work being LENS (Li & Qiu, 2023), which depends on LLM output probabilities, is expensive in terms of number of LLM calls and is unsuitable for black-box models. We propose a principled novel static exemplar selection method grounded in gap-index based stochastic linear bandits that works for both black-box and open LLMs.

**Identification of Top-$m$ Arms in Stochastic Linear Bandits.** The top-$m$ arms identification problem aims to estimate a subset of $m$ arms with the highest means. Various methods have been proposed for top-$m$ arm identification in both fixed-confidence (Kalyanakrishnan et al., 2012) and fixed-budget settings (Bubeck et al., 2013). In this paper, we focus on the fixed-confidence setting, where the error probability in estimating the top-$m$ arms should be smaller than a predefined parameter $\delta \in (0, 1)$. Adaptive sampling algorithms like UGapE (Gabillon et al., 2012) and LUCB (Kalyanakrishnan et al., 2012), along with uniform sampling methods (Kaufmann & Kalyanakrishnan, 2013; Chen et al., 2017), have been introduced for the fixed confidence setup, but they lack efficiency in terms of sample complexity. While efficient adaptive sampling methods for linear bandits, such as Fiez et al. (2019), RAGE Zhang et al. (2023a), LTS Jedra & Proutiere (2020), PEPS Li et al. (2023), LinGapE (Xu et al., 2017) and LinGame Degenne et al. (2020), have been proposed, they primarily address best-arm identification ($m = 1$). To the best of our knowledge, GIFA (Réda et al., 2021) was the first unified framework for efficient top-$m$ arm identification with low sample complexity. However, algorithms implemented in GIFA framework require large number of gap-index computations and comparisons, leading to high sample complexity. This is due to its challenger arm sampling mechanism, which considers the complement of the current top-$m$ estimate as the challenger set, resulting in a large search space. In this work, we propose a novel and efficient algorithm with applications to exemplar selection for complex reasoning in LLMs.

## 3 CASE: CHALLENGER ARM SAMPLING-BASED EXPLORATION FOR EXEMPLAR-SUBSET SELECTION

In-context learning (ICL) leverages LLMs to acquire task-specific knowledge from just a few examples, typically structured as *input-rationale-output* triplets, without updating the model's parameter. The rationales serve to elucidate the reasoning process behind the input-output pair, fostering reasoning capabilities within LLMs. However, due to the financial and computational costs associated with processing large contexts, providing all training examples is impractical. Therefore, a key challenge

lies in the efficient and optimal selection of exemplars, particularly in a black-box setting where access to the model's parameters or confidence estimates is unavailable.

To address this challenge, we propose a novel algorithm for exemplar subset selection. First, within the in-context learning paradigm, the task of constructing an optimal prompt is framed as a *multiple exemplar subset selection* problem (section 3.1). In section 3.2, we model this as a top-$m$ arm selection problem in stochastic linear bandits (Kalyanakrishnan & Stone, 2010; Réda et al., 2021), where a linear function is employed to approximate the loss of each arm (prompt). Section 3.3, introduces a new algorithm based on *selective exploration*—conceptually related to SETC (Lattimore & Szepesvári, 2020)), which samples and stores the "challenger arms" (arms that have the potential to belong to the top-$m$ set (Réda et al., 2021)). This approach significantly reduces the computational overhead by avoiding the need to evaluate all possible arms in every iteration.

### 3.1 MULTIPLE EXEMPLAR-SUBSET SELECTION FOR IN-CONTEXT REASONING

In-context learning (ICL) allows LLMs to be used for supervised learning tasks by leveraging a few demonstration examples (called exemplars) of the task without training or parameter fine-tuning. Unlike traditional supervised learning, where training examples include the input features $u$ and output labels $v$, exemplars sometime include chain of thought demonstrations, which are not available for test examples. For simplicity, we include them in $u$ for the training exemplars. Let $\mathcal{X} = \{u_i, v_i\}_{i=1}^n$ be the set of all $n$ potential training exemplars and $(u_{test}, v_{test})$ denote a test input features and desired output. A prompt $P$ is constructed from a subset $S \subseteq \mathcal{X}$ of $k$ exemplars, which can be used for prediction of $u_{test}$. Hence, $P = [S, u_{test}] = [(u_{i_1}, v_{i_1}), ..., (u_{i_k}, v_{i_k}), u_{test}]$. The prompt $P$ is then passed to a response generator function $f$ which uses a decoding mechanism $\mathcal{G}$ to sample responses from an LLM. Hence, $f(P) = \mathcal{G}(\mathbb{P}_{LLM}(r|P))$. The final step is a post-processing $\delta$ applied to the LLM-generated response $f(P)$, in order to extract the task-specific output $\hat{v}_{test}$. Commonly used post-processing strategies include regular-expression matching ($\delta_{regex}$) and self-consistency ($\delta_{SC}$) Wang et al. (2023b) based strategies.

*In-context reasoning* (ICR) uses ICL to learn prompts for tasks involving reasoning with natural language, e.g. numerical reasoning based Question Answering (QA) (Chen et al., 2022b; Lu et al., 2023b), commonsense reasoning based QA (Geva et al., 2021b) etc. Due to the inherent difficulty of ICR, prompt generator $\pi$ which uses multiple high-scoring subsets $S_1, ..., S_m$ and a test input $u_{test}$ to generate a prompt $P$ are used. Such prompts $P = \pi(S_1, ..., S_m, u_{test})$ are expected to improve the overall performance or robustness of the underlying ICR task. Examples of prompt generators are a similarity-based prompt generator $\pi_{KNN}$ and a diversity-based prompt generator $\pi_{MMR}$ (discussed in section 4). Hence, the entire output generation process can be described as:

$$\hat{v}_{test} = \delta(f(P(U(\mathcal{X})))), \text{ where } P(\mathcal{X}) = \pi(U(\mathcal{X}), u_{test}), \text{ where } U(\mathcal{X}) = (S_1, ..., S_m)(\mathcal{X}) \quad (1)$$

Here $U(\mathcal{X}) = (S_1, ..., S_m)(\mathcal{X}))$ a set of $m$ subsets of the training set $\mathcal{X}$. Let $\mathcal{V}$ be the set of $n'$ validation examples $\{u'_i, v'_i\}_{i=1}^{n'}$, and define the validation accuracy for a prompt $P(U(\mathcal{X}))$ as: $A(P(U(\mathcal{X})), \mathcal{V}) = \frac{1}{n'} \sum_{i=1}^{n'} \mathbb{1}(v'_i = \delta(f(\pi(U(\mathcal{X}), u'_i))))$. For ICR, we are interested in finding a set $U(\mathcal{X})$ of $m$-subsets of $\mathcal{X}$ such that the corresponding prompt $P(U(\mathcal{X}))$ generated by the prompt generator $\pi$ maximizes the total validation accuracy.

$$U^*(\mathcal{X}) = \underset{U \in \mathcal{S}(\mathcal{X})^m}{\arg\max} A(P(U), \mathcal{V}) \text{ where } \mathcal{S}(\mathcal{X}) \text{contains all } k\text{-subsets of } \mathcal{X} \quad (2)$$

We call this as *multiple exemplar-subset selection* (MESS) formulation for finding the optimal prompt.

### 3.2 TOP-$m$ ARM SELECTION FORMULATION FOR PROMPT GENERATION

The MESS problem defined above is a discrete optimization problem over an exponentially large search space $\mathcal{S}(\mathcal{X})^m$. Additionally, the function $A$ is computationally expensive and non-differentiable due to the black-box (oracle) access to the LLM response generator function $f$. Hence, naive or heuristic search-based algorithms for solving MESS are computationally infeasible in this setting. Further, due to the varied nature of the different prompt generators ($\pi$), the accuracy of the actual generated prompt is difficult to optimize. Hence, we optimize the average accuracy of the prompts generated by each of the $m$-subsets. Let $a \in \{0, 1\}^n$ such that $\|a\|_1 = k$ denote the encoding for an exemplar subset of size $k$, which also denote a prompt $P = \pi_r(a)$. Here, $\pi_r$ is a prompt

generator that takes a random ordering of the exemplars in the subset denoted by $a$. Considering each $a_i \in S$ as the *arms* and $A(\pi_r(a), \mathcal{V})$ as the *reward* for arm $a$, we use multi-armed bandit (MAB)-based algorithms (Lattimore & Szepesvári, 2020) for sample-efficient exploration of the arms space. Particularly, we pose the modified MESS problem as a top-$m$ arm identification problem (Kalyanakrishnan & Stone, 2010; Bubeck et al., 2009), also called the pure exploration problem, and design an algorithm that iteratively samples the reward for an arm $a(t)$ at each iteration $t$.

We further assume that the observed reward $A(\pi_r(a(t)), \mathcal{V})$ can be modeled as a linear function $\rho(a(t))$ of the features of arm $a(t)$. Intuitively, given an arm $a(t)$, the reward scoring function $\rho(a(t))$ should be a function of the similarity between the texts of the selected exemplar, $u_i$ such that $a_i = 1$, and the validation exemplar $u_j \in \mathcal{V}$. In this work, we use a normalized BERT-based similarity score, $\sigma_{ij} = \frac{\phi(u_i)^T \phi(u'_j)}{\|\phi(u_i)\|\|\phi(u'_j)\|}$, where $\phi(u)$ is the sentence encoding vector obtained from a pre-trained transformer model, e.g. SentenceBERT. Let, $\alpha_i, i = 1, ..., n$ denote the $i$-th training exemplar's coefficient in the reward scoring function. Our linear model for the observed reward of an arm $a(t)$ at the $t$-th iteration can be written as:

$$\rho(\vec{\alpha}, a(t)) = \frac{1}{n'} \sum_{j=1}^{n'} \sum_{i=1}^{n} \alpha_i a(t)_i \sigma_{ij} + \eta_t = \sum_{i=1}^{n} \alpha_i a(t)_i x_i + \eta_t, \text{ where } x_i = \frac{1}{n'} \sum_{j=1}^{n'} \sigma_{i,j} \quad (3)$$

and $\eta_t$ is a subgaussian noise, i.e. $\mathbb{E}[e^{\lambda \eta_t}] \leq \exp(\lambda^2 \xi^2/2)$, for some variance $\xi^2$. Under this stochastic linear bandits assumption, the problem of identifying top-$m$ arms can solved using the Gap-index based algorithms (Xu et al., 2018b; Réda et al., 2021), which were unified under the GIFA framework (Réda et al., 2021). The gap-index between any two arms $i, j$, $B_t(i, j)$, is defined as the confidence-enhanced gap between their estimated mean rewards $\hat{\rho}_t$. Hence, $B_t(i, j) = \hat{\rho}_t(i) - \hat{\rho}_t(j) + W_t(i, j)$, where the confidence term is defined as: $W_t(i, j) = \left[ \sqrt{2 \ln \left( \frac{1}{\delta} \right) + N \ln \left( 1 + \frac{(t+1)L^2}{\lambda^2 N} \right)} + \frac{\sqrt{\lambda}}{\sigma} S \right] (\|x_i\|_{\hat{\Sigma}_t^\lambda} + \|x_j\|_{\hat{\Sigma}_t^\lambda})$, where $S$ and $L$ are constants, $N$ is the total number of arms pulled, and $\hat{\Sigma}_t^\lambda = \sigma^2 (\hat{V}_t)^{-1}$ ($\hat{V}_t$ is the design matrix defined below).

GIFA algorithms maintain a set of estimated $m$-best arms $U_t$, In each iteration $t$, the most ambiguous arm from $U_t$, say $b_t = \arg\max_{b \in U_t} \max_{a \in U_t^c} B_t(a, b)$, and the most ambiguous arm from $U_t^c$ (called the *challenger* arm), denoted by $c_t = \arg\max_{c \in U_t^c} B_t(c, b_t)$, are computed. The arm with the highest variance between $b_t$ and $c_t$ is pulled, and the model parameters are updated. The design matrix $\hat{V}_{t+1}$ is defined as: $\hat{V}_{t+1} = \lambda I_n + \sum_{a \in S} N_a x_a x_a^T$, where $N_a$ is the number of times arm $a$ was pulled. The updated parameters $\hat{\alpha}_{t+1}$ are computed as: $\hat{\alpha}_{t+1} = (\hat{V}_{t+1})^{-1} (\sum_{l=1}^{t+1} r_l x_{a_l})$, where $r_l$ is the reward received by computing the accuracy of the prompt generated by the current arm. The main problem with the GIFA framework is that the total number of arms $|S|$ is exponentially large. Hence, the computation of most ambiguous arm $b_t$ and challenger arm $c_t$ is infeasible. Next, we propose a new scheme for mitigating this problem.

### 3.3 CASE: Challenger-arm sampling-based top-$m$ arm selection algorithm

Implementing gap-index-based schemes for top-$m$ arm identification, for settings with exponentially large number of arms is infeasible. The key problem is to identify the most ambiguous arms in each iteration. We propose to mitigate this problem using: (a) identify a low-regret subset $N_t$ of $m'$ *next-best arms* after $U_t$, and (b) use a GIFA-based algorithm to identify the top-$m$ arms from the set $U_t \cup N_t$. The problem of combinatorial blowup of arms in the linear bandit setting has been sparsely studied, with selective exploration as one of the strategies (e.g. Algorithm 13, Chapter 23 of Lattimore & Szepesvári (2020)). Since it is impractical to explore all arms, the selective exploration scheme uniformly samples arms from the unexplored set and then selects the highest-scoring arms according to the current model. It then pulls the selected arms to update the model parameters.

Algorithm 1 describes the proposed *challenger-arm sampling* based exploration technique, called CASE. The set of top-m subsets (arms) $U_0$ is initialized to a random set sampled from $S$. In lines $11 - 14$, we compute the updated $U_t$ by moving the highest scoring arm from $N_{t-1}$ if its score is higher than that of the lowest scoring arm in $U_{t-1}$, which is then moved to $N_{t-1}$. Using the selective exploration idea, CASE uniformly samples $m'$ arms from $(U_t \cup N_{t-1})^c$, to generate the set $M_t$, and then selects the top-$m'$ arms from $M_t \cup N_{t-1}$ to generate the updated $N_t$. $N_t \cup U_t$ is

---

**Algorithm 1:** CASE: **C**hallenger **A**rm **S**ampling for **E**xemplar selection

---

1 **Input:** $\mathcal{X}$: set of all training exemplars, $k$: prompt size, $\mathcal{S}$ : all $k$-subsets of $\mathcal{X}$, $a \in \mathcal{S}$: an arm or $k$-subset

2

3 **Define:**    $U_t$: set of currently estimated top-$m$ arms.

4              $N_t$: set of currently estimated next best-$m'$ arms.

5              $b_t$: the most ambiguous arm from $U_t$

6              $s_t$: the most ambiguous sampled arm from $N_t$

7

8 **Initialize:** $U_0 \leftarrow$ set of random $m$ arms from $\mathcal{S}$,  $t \leftarrow 1$,  $\vec{\alpha}_1 \leftarrow \mathcal{N}(0,1)$

9 **while** $B_t(s_t, b_t) \leq \epsilon$ **do**

10    *Construct $U_t$ by replacing $n_t \in U_{t-1}$ with a potentially better arm $c_t \in N_{t-1}$*

11    $n_t = \arg\min_{a \in U_{t-1}} \hat{\rho}_t(a)$; $c_t = \arg\max_{a \in N_{t-1}} \hat{\rho}_t(a)$

12    **if** $\hat{\rho}_t(c_t) \geq \hat{\rho}_t(n_t)$ **then**

13       |  $U_t, N_t \leftarrow \text{swap}(n_t, c_t)$ from $U_{t-1}, N_{t-1}$

14    **end**

15    $M_t \leftarrow s' \sim_{m'} (U_t \cup N_{t-1})^c$           $\triangleright$ Randomly sample from $(U_t \cup N_{t-1})^c$

16    $N_t \leftarrow \text{top}_{m'}(M_t \cup N_{t-1}; \hat{\rho}_{(t)})$           $\triangleright$ Updated $N_t$ from $N_{t-1}$ and $M_t$

17    *Compute the revised most ambiguous arms for detecting convergence*

18    $b_{t+1} = \arg\max_{b \in U_t} \max_{a \in N_t} [B_t(a, b) = \hat{\rho}_t(a) - \hat{\rho}_t(b) + W_t(a, b)]$

19    $s_{t+1} = \arg\max_{s \in N_t} [B_t(s, b_{t+1}) = \hat{\rho}_t(s) - \hat{\rho}_t(b_t) + W_t(s, b_{t+1})]$

20    *Pull selected arm, receive reward, and update model parameters*

21    $a_{t+1} \leftarrow \text{selection\_rule}(U_t, N_t)$

22    $r_{t+1} = A(\pi_r(a(t)), \mathcal{V})$

23    $\hat{V}_{t+1} = \lambda I_n + \sum_{a \in \mathcal{S}} N_a x_a x_a^T$           $\triangleright$ $\lambda$ regularized design matrix

24    $\hat{\alpha}_{t+1} = (\hat{V}_{t+1})^{-1}(\sum_{l=1}^{t+1} r_l x_{a_l})$           $\triangleright$ Least-squares estimate

25    $t \leftarrow t + 1$

26 **end**

27 **Output:** $U_T$: Set of $m$ arms from $K$ which have the highest reward

---

the high-reward selected set, from which we explore using the selection rule. We use the greedy selection rule proposed in (Réda et al., 2021), where we select the arm that minimizes the variance between $s_t$ and $b_t$: $a^* = \arg\min_{a \in N_t \cup U_t} ||x_{b_t} - x_{s_t}||_{(\hat{V}_{t-1} + x_a x_a^T)^{-1}}$. We call the LLM (line 22 in Algorithm 1) after the selection rule, where an arm (i.e., a set of exemplars) is sampled. Specifically, $A(\pi_r(a(t)), \mathcal{V})$ denotes to the computation of accuracy on the validation subset, which requires LLM to generate predictions. This ensures that the LLM's reasoning process and output generation are explicitly integrated into our algorithm. By doing so, we effectively leverage the gap-index-based multi-arm bandit framework to optimize exemplar selection for complex reasoning tasks. Steps 18 and 19 in algorithm 1 compute the revised most ambiguous arms $b_t \in U_t$ and $s_t \in N_t$. $s_t$ is *sampled challenger* arm selected from the selected set of next best arms $N_t$ using the gap index $B_t(a, b) = \hat{\rho}_t(a) - \hat{\rho}_t(b) + W_t(a, b)$. Finally, the revised parameters $\hat{\alpha}_{t+1}$ are computed using the revised design matrix $\hat{V}_{t+1}$ and the least squares estimation formulae described in lines 23 and 24 of algorithm 1. Note that, $\hat{V}_{t+1}$ can be computed from $\hat{V}_t$ using the incremental update formula $\hat{V}_{t+1} = \hat{V}_t + x_{a_{t+1}} x_{a_{t+1}}^T$. Hence, $(\hat{V}_{t+1})^{-1}$ can be calculated in $O(n^2)$ imte using the Sherman-Morrison formula. We stop the updates when the convergence criteria for switching the arms between $U_t$ and $N_t$ have been achieved, i.e. $B_t(s_t, b_t) \leq \epsilon$. The time complexity for each iteration of our algorithm is $O(mm' + n^2 + \text{LLM\_inference\_time})$ which is due to lines 18, 21 and 24. Next, we discuss some theoretical results related to our method.

### 3.4 SAMPLE COMPLEXITY BOUNDS FOR THE TOP-$m$ SELECTION ALGORITHM

The ability of the proposed algorithm to identify the top-$m$ arm and correctly estimate the model parameters $\alpha$ rests on two arguments. Firstly, the selective exploration strategy results in a low regret set of arms $U_t \cup N_T$. Specifically, we assume the average regret (total regret / #iterations) of the set $U_T \cup N_T$ to upper bounded by $\epsilon$. While we postpone a rigorous derivation of the regret bound for CASE to a later study, we justify our assumption by using the SETC Algorithm (Algo 13 in (Lattimore & Szepesvári, 2020)), which follows a similar uniform exploration and commitment strategy in the linear bandit setting. SETC has a regret bound of $O(d\sqrt{n \log(n)})$ where $n$ is the

number of time steps. Hence, the algorithm will achieve an average regret of at most $\epsilon$ after at most $\exp(W(\frac{\epsilon^2}{C^2 d^2}))$ timesteps, where $W$ is Lambert's function, and $C$ is the constant for the regret bound.

Secondly, once a low regret $U_T \cup N_T$ set is achieved, the gap-index-based algorithm correctly identifies the top-$m$ arms from the set $U_T \cup N_T$. Following (Réda et al., 2021), we obtain a high probability $(1 - \delta)$ upper bound for the sample complexity of CASE on the event $\mathcal{E} \triangleq \bigcap_{t>0} \bigcap_{i,j \in [K]} \Big( \rho_i - \rho_j \in [-B_t(j,i), B_t(i,j)] \Big)$, Let $\mathcal{S}_m^\star$ be the true set of top-$m$ arms. We define the true gap of an arm $i$ as $\Delta(i) \triangleq \rho(i) - \rho(m+1)$ if $i \in \mathcal{S}_m^\star$, $\rho(m) - \rho(i)$ otherwise ($\Delta(i) \geq 0$ for any $i \in [K]$).

**Theorem 1.** *For* CASE*, on event $\mathcal{E}$ on which the algorithm is $(\varepsilon, m, \delta)$-PAC, stopping time $\tau_\delta$ satisfies $\tau_\delta \leq \inf\{u \in \rho^{*+} : u > 1 + H^\varepsilon(\mu) C_{\delta,u}^2 + \mathcal{O}(K)\}$, where, for algorithm with the largest variance selection rule[2] : $H^\varepsilon(\mu) \triangleq 4\sigma^2 \sum_{a \in [K]} \max\left(\varepsilon, \frac{\varepsilon + \Delta_a}{3}\right)^{-2}$,*

Above theorem essentially adopts the result in Theorem 2 from Réda et al. (2021) to the setting where we restrict ourselves to arms in $U_T \cup N_T$. It mentions that the $\epsilon$-optimal top-m arms from $U_T \cup N_T$ are present in $U_T$ with prob. $1 - \delta$, if $T > \tau_\delta$. $K$ is the size of $U_T \cup N_T$.

**Proof Overview**: The proof builds upon the proofs for classical Top-$m$ linear bandits, LinGapE Xu et al. (2018a) and LinGIFA Réda et al. (2021) while additionally accounting for the challenger arms shortlist in $N_t$ proposed in this work. To prove it, one of the key components is the following lemma, which holds for any gap indices of the form $B_t(i,j) \triangleq \hat{\rho}_t(i) - \hat{\rho}_t(j) + W_t(i,j)$ for $i, j \in [K]^2$.

**Lemma 1.** *On the event $\mathcal{E}$, for all $t > 0$, $B_t(s_t, b_t)(t) \leq \min(-(\Delta(b_t) \vee \Delta(s_t)) + 2W_t(b_t, s_t), 0) + W_t(b_t, s_t)$; where $a \vee b = max(a, b)$.*
The proof for Lemma 1 is provided in Appendix A.1. In summary, Theorem 1 and Lemma 1 together provide an upper bound on the expected number of arm pulls required by the algorithm, which translates to the number of LLM calls needed when applying CASE to complex reasoning tasks.

## 4 EXPERIMENTS AND RESULTS

We aim to address the following research questions:
**RQ I.** How computationally efficient is CASE compared to other stochastic linear bandit approaches?
**RQ II.** Can CASE, a task-level exemplar selection method, achieve competitive performance across diverse tasks compared to state-of-the-art exemplar selection methods?
**RQ III.** Can exemplars selected using smaller LLMs be effectively reused for larger LLMs?
**RQ IV.** How efficient is CASE for exemplar selection compared to state-of-the-art methods?

### 4.1 EXPERIMENTAL SETUP

**Synthetic Experiments** For synthetic experiments, we adopt a setup similar to Réda et al. (2021) and present results on simulated data. We set $\sigma = 0.5$, $\epsilon = 0$, and $\delta = 0.05$ across all experiments. Each experiment is conducted over 500 simulations. We explore various numbers of arms $K \in [7, 10, 20]$, and set the number of top arms to be identified with the highest means as $m = 3$. We choose a challenger set $N_t$ of size 3. The feature dimension is fixed at 3 for all synthetic experiments.

#### SETUP FOR TASK LEVEL EXEMPLAR SELECTION FOR COMPLEX REASONING

**Datasets and Metrics:** We evaluate on well-known datasets that require complex reasoning. For numerical reasoning, we use GSM8K and AquaRAT; for commonsense reasoning, we use StrategyQA; and for tabular and numerical reasoning, we use FinQA and TabMWP. Detailed descriptions of the datasets are provided in Appendix C. We report performance using the official metrics: Exact Match (EM) and Cover-EM Rosset et al. (2021) for the respective datasets.

**Hyperparameters (LLMs):** For LLMs, we set the temperature to 0.3 to reduce randomness in the generated outputs. To reduce repetition, we apply a presence penalty of 0.6 and a frequency penalty of 0.8. The $max\_length$ for generation is set to 900.

---

[2]or pulling both arms in $\{b_t, c_t\}$ at time $t$

Table 1: Results across datasets. Percentage improvements are reported over LENS (Li & Qiu, 2023). † indicates statistical significance using t-test over LENS at 0.05 level and ‡ at 0.01 level.

| Method | GSM8K | AquaRat | TabMWP | FinQA | StrategyQA |
|---|---|---|---|---|---|
| | GPT-3.5-turbo | | | | |
| **Instance level** | | | | | |
| KNN (Rubin et al., 2022) | 53.45 | 51.96 | 77.07 | 51.52 | 81.83 |
| KNN (S-BERT) (Rubin et al., 2022) | 53.07 | 52.75 | 77.95 | 52.65 | 81.83 |
| MMR (Ye et al., 2023b) | 54.36 | 51.18 | 77.32 | 49.87 | 82.86 |
| KNN+SC (Wang et al., 2023b) | 80.21 | 62.59 | 83.08 | 54.49 | 83.88 |
| MMR+SC (Wang et al., 2023b) | 78.01 | 59.45 | 81.36 | 50.74 | 83.88 |
| PromptPG (Lu et al., 2023a) | - | - | 68.23 | 53.56 | - |
| **Task level** | | | | | |
| Zero-Shot COT (Kojima et al., 2023) | 67.02 | 49.60 | 57.10 | 47.51 | 59.75 |
| Manual Few-Shot COT (Wei et al., 2023) | 73.46 | 44.88 | 71.22 | 52.22 | 73.06 |
| Auto-COT (Zhang et al., 2023b) | 62.62 | 43.31 | - | - | 71.20 |
| Complex-COT (Fu et al., 2023) | 71.04 | 51.18 | - | - | 71.63 |
| Random | 67.79 | 49.80 | 55.89 | 53.70 | 81.02 |
| PS+ (Wang et al., 2023a) | 59.30 | 46.00 | - | - | - |
| GraphCut (Iyer & Bilmes, 2013) | 66.19 | 47.24 | 60.45 | 52.31 | 80.00 |
| FacilityLocation (Iyer & Bilmes, 2013) | 68.61 | 48.43 | 67.66 | 36.79 | 81.63 |
| LENS (Li & Qiu, 2023) | 69.37 | 48.82 | 77.27 | 54.75 | 79.79 |
| LENS+SC (Li & Qiu, 2023) | 79.37 | 57.87 | 80.68 | 60.06 | 82.24 |
| LENS+KNN+SC (Li & Qiu, 2023) | 80.06 | 57.87 | 79.79 | 60.68 | 83.46 |
| LENS+MMR+SC (Li & Qiu, 2023) | 80.36 | 59.44 | 80.39 | 60.94 | 82.45 |
| **Our Approach** | | | | | |
| **CASE** | 79.91(▲15.19%)‡ | 54.72(▲12.09%) | 83.42(▲7.96%)‡ | 59.72(▲9.08%)† | 84.49(▲5.89%)† |
| **CASE+SC** | 86.96(▲25.36%)‡ | 62.20(▲27.41%)‡ | 84.91(▲9.89%)‡ | 63.64(▲16.24%)‡ | 87.55(▲9.73%)‡ |
| **CASE+KNN+SC** | **87.49**(▲26.12%)‡ | **64.17**(▲31.44%)‡ | **86.23**(▲11.60%)‡ | **64.25**(▲17.35%)‡ | 85.92(▲7.68%)‡ |
| **CASE+MMR+SC** | 85.60 (▲23.40%)‡ | 62.60(▲28.23%)‡ | 85.91(▲11.18%)‡ | 63.47(▲15.93%)‡ | 84.69(▲6.14%)‡ |
| | GPT-4o-mini | | | | |
| LENS (Li & Qiu, 2023) | 76.19 | 64.56 | 86.34 | 69.31 | 92.85 |
| **CASE** | **91.13** | **73.23** | **89.73** | **72.89** | **95.92** |

**Hyperparameters (CASE):** In CASE we begin by clustering the training set into 5 clusters. We then form the set of exemplar subsets ($\mathcal{S}$), by sampling one exemplar from each cluster *with replacement*. This approach allows us to explore various combinations of exemplars. Note that the training set to form $\mathcal{S}$ are the same across baselines for a fair comparison. Our main results are reported for $|\mathcal{S}| = 100$. We set $m = |U_t| = 10$ to identify the top scoring subsets (arms) and choose a challenger set $N_t$ of size 5. We set the number of validation examples $\mathcal{V}$ to, 20 and $\epsilon = 0.1$ (stopping criterion).

**Baselines:** We compare with zero-shot Kojima et al. (2023) and manual few-shot Wei et al. (2023) settings with COT rationales to highlight the importance of careful exemplar selection. Additionally, we compare against instance-level exemplar selection methods, such as MMR Ye et al. (2023b) and KNN Rubin et al. (2022), which use diversity and similarity-based measures to select exemplars for each test example. For fair comparison, we set the number of exemplars to 5 and configure MMR with $\lambda = 0.5$ to balance similarity and diversity. We also evaluate against coreset selection methods like Graphcut and Facility Location Iyer & Bilmes (2013). Finally, we compare with LENS Li & Qiu (2023), a state-of-the-art task-level exemplar selection approach that is closely aligned with our work.

**CASE Hybrid Variants:** We also propose hybrid variants of CASE, where we select instance-level exemplars subset from the top-$m$ subsets identified by CASE using KNN or MMR, thereby reducing the search space. While KNN and MMR typically select individual exemplars for each test instance, our hybrid approach scores entire subsets by aggregating the similarity scores of each exemplar within the subset to the test instance, thereby preserving the interactions between exemplars.

## 4.2 PERFORMANCE OF CASE COMPARED TO EXISTING GAP-INDEX-BASED APPROACHES

To address **RQ1**, we conduct synthetic experiments following the setup in Section 4.1. We evaluate metrics such as average runtime, the average number of comparisons for gap index computation, for multi-armed bandit (MAB) approaches including LinGapE, LinGIFA, and CASE. The results for $K = 20$, $m = 3$, and $N = 3$ are shown in Figure 2. Other synthetic experiments are presented in Appendix H. In Figure 2(a), we observe that CASE significantly reduces the number of comparisons required for gap index computations compared to state-of-the-art gap-index-based MAB approaches. CASE requires significantly fewer comparisons than LinGapE. This improvement is largely due to the novel and principled challenger arm sampling strategy, resulting in efficient gap index computations. The efficiency gains stem from the fact that $|N_t| < |U_t^c|$, where existing gap-index frameworks use the

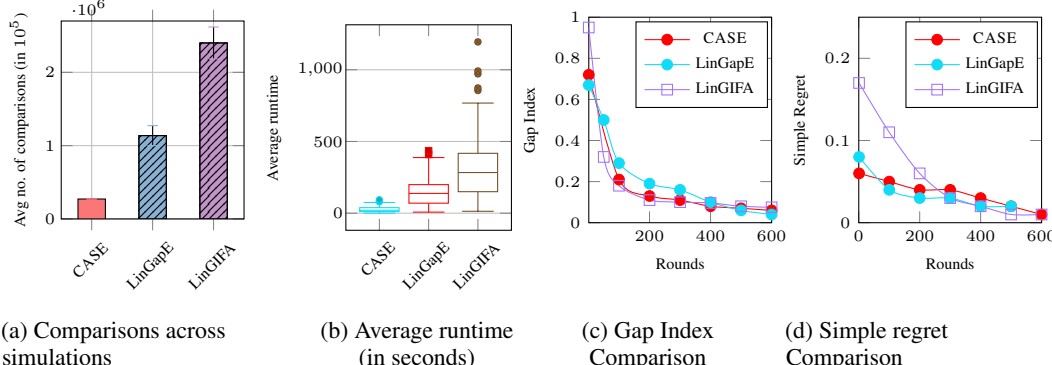

(a) Comparisons across simulations     (b) Average runtime (in seconds)     (c) Gap Index Comparison     (d) Simple regret Comparison

Figure 2: Top-$m$ arm identification by CASE, LinGIFA and LinGapE for K=20, m=3, N=3. (a) Average number of comparisons across simulations (b) Average runtime (in seconds) (c) Gap Index ($B_t(s_t, b_t)$) comparison and (d) Simple regret comparison for each round across simulations

entire $U_t^c$ to select challenger arms, leading to a larger number of computations. In contrast, CASE, prunes the space of possible challenger arms ($N_t$), leading to significant efficiency improvements. From Figure 2(b), we also observe that runtime of CASE is lower when compared to LinGapE (about **5.6x**) and LinGIFA (about **12x**) due to lower number of arm comparisons and gap-index computations. In Figures 2(c) and 2(d), we analyze the gap index ($B_t(s_t, b_t)$) and simple regret across rounds (averaged across simulations) and observe that they approach 0 as the number of rounds increases. This demonstrates that, CASE samples good arms with our shortlist of $N_t$ serving as a good approximation of challenger arms. CASE converges with much lower gap index computations and has lower runtime compared to existing state-of-the-art gap index algorithms. We also observed that all algorithms have similar average sample complexity with negligible differences.

### 4.3 Performance Comparison on Task-Level Exemplar Selection

To address **RQ II**, we compare CASE and its variants against state-of-the-art task-level and instance-level exemplar selection methods. Table 1 shows that CASE consistently outperforms the random, as well as zero-shot and manual Few-Shot COT Wei et al. (2023) methods, which rely on random or hand-picked samples without accounting for the interactions between exemplars and task-level performance. Furthermore, classical coreset selection methods like Graph Cut and Facility Location, which evaluate a subset's informativeness and diversity, perform worse or on par with random selection. This is largely because these coreset methods were not designed for the ICL paradigm. We also observe that CASE outperforms dynamic selection methods like KNN, PromptPG, and MMR. Beyond the efficiency gained during inference, task-level exemplars demonstrate greater robustness (see **Appendix** E) compared to dynamic methods, which selects exemplars for each test instance, introducing more variance. The independent selection of exemplars in dynamic methods may result in negative interactions or skill redundancy, where lexically different exemplars may represent similar skills, limiting their utility in reasoning tasks. To further support the claim that CASE selects more representative task-level exemplars, we conduct a *qualitative analysis* comparing exemplars chosen by LENS and CASE (see **Appendix G**). We find that CASE consistently selects exemplars with diverse skills required for solving tasks, whereas LENS selects exemplars with redundant skills.

### 4.4 Exemplar Reuse and Selection Efficiency

To address **RQ III**, we evaluate the performance of gpt-3.5-turbo using exemplars selected by smaller models, such as Llama2-7b (see Figure 3(a)) and Mistral-7b. In Table 1 we present the results of exemplar reuse from Mistral-7b for CASE. We observe that exemplars selected by CASE using smaller LLMs perform well with larger LLMs, promoting both exemplar reuse and efficiency.

To address **RQ IV**, we compare the number of LLM calls made by CASE and LENS (shown in Figure 3(b)). We observe that CASE reduces LLM calls significantly compared to LENS (about **6x to 10.5x**) and also reduces the exemplar selection time (shown in Figure 3(c)). This difference arises because LENS iteratively eliminates each training example by computing an informativeness and

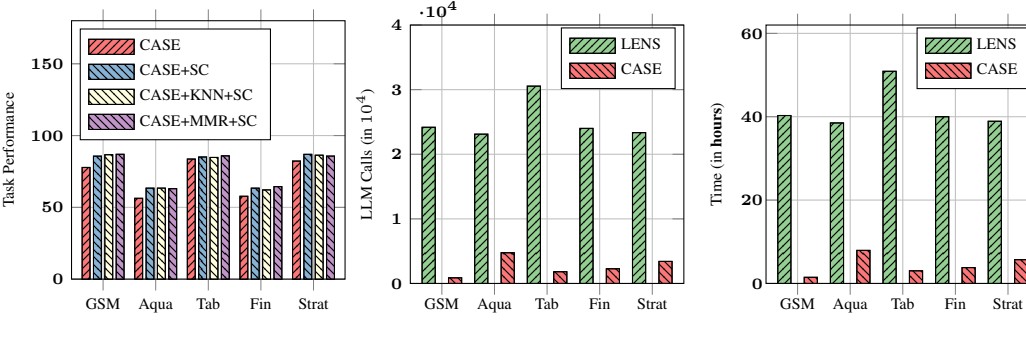

(a) Reuse, Llama2 to gpt-3.5-turbo.     (b) LLM calls LENS vs CASE     (c) Runtime LENS vs CASE

Figure 3: (a) Exemplar Reuse for LLMs, (b-c) Efficiency of CASE compared to LENS.

diversity measure based on LLM confidence estimates, resulting in an LLM call for each training instance over multiple rounds. In contrast, CASE directly scores exemplar subsets, and its novel challenger set sampling mechanism dramatically reduces the number of subsets (arms) that need to be explored and evaluated. Consequently, CASE is more efficient than even state-of-the-art gap-index-based MAB algorithms like LinGIFA and LinGapE, as demonstrated in Figure 2.

## 4.5 ABLATION STUDIES

We conduct several ablation studies, including one-time sampling of all exemplar subsets and a variant of CASE without exploration, to highlight the need for an exploration-based MAB approach, as shown in Table 2. In the one-time sampling approach, we sample each subset once, evaluate them on the validation set, and select the subset with the lowest validation loss. The test set inference results using the selected subset are presented in Table 2. We observe that one-time sampling underperforms compared to CASE since it evaluates all arms only once and lacks sufficient information to confidently identify the best arms. This method tends to overfit the validation set, providing only a baseline understanding of the exemplar subsets (arms) and failing to lead to optimal selection without incorporating exploration or exploitation mechanisms.

Furthermore, we conduct an ablation where we fit the linear model once on a set of $\mathcal{S}$ subsets (where $|\mathcal{S}|=K$ (number of arms)=100) (as shown in Table 2) and select the subset with the highest mean for test set inference. Unlike CASE, which models the top-$m$ subsets in each round, this ablation fits the linear model once for all subsets. Since, this ablation is devoid of exploration mechanism it does not result in confident estimation of top-m arms resulting in suboptimal performance.

Table 2: Ablation studies: one-time sampling, w/o exploration vs proposed exploration (CASE).

| Datasets | GSM | Aqua | Tab | Fin | Strat |
|---|---|---|---|---|---|
| One-time sampling | 76.72 | 50.39 | 81.23 | 54.14 | 80.20 |
| CASE (-exploration) (Mistral) | 76.57 | 47.64 | 77.17 | 45.95 | 80.00 |
| **CASE (Llama)** | **77.79** | **56.30** | **83.65** | **57.72** | **82.24** |
| **CASE (Mistral)** | **79.91** | **54.72** | **83.42** | **59.72** | **84.49** |

## 5 CONCLUSION

In this work, we propose an efficient task-level exemplar subset selection method that identifies highly informative exemplar subsets. Our approach significantly reduces resource consumption by minimizing the number of LLM calls, offering a clear advantage over existing state-of-the-art exemplar selection methods. Additionally, we find that exemplars selected using smaller LLMs can be effectively reused for larger models. CASE not only outperforms existing static exemplar selection methods but is also superior to, dynamic selection methods. In the future, we plan to derive a rigorous regret bound for CASE, explore the mechanisms of ICL and the role of exemplars in tasks requiring complex reasoning.

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

# A PROOFS

## A.1 PROOF OF LEMMA 1

*Proof.* To prove the above case, we first introduce the following property. We primarily follow the proof structure of GIFA framework (Réda et al., 2021) with some modifications required for CASE due to the shortlist $N_t$ and our swapping rule to compute $U_t$.

Let $\mathcal{S}_m^\star$ be the true set of top-$m$ arms and $(S_m^*)^c$ denote the true set remaining worst arms.

**Property 1**: For $b_t \in U_t$ and $s_t \in N_t$ it holds that $\hat{\rho}_t(b_t) \geq \hat{\rho}_t(s_t)$. Hence, it follows that $B_t(s_t, b_t) = \hat{\Delta}_t(s_t, b_t) + W_t(b_t, s_t) \leq W_t(b_t, s_t)$ as $\hat{\Delta}_t(s_t, b_t) < 0$ From property 1, we can establish that $B_t(s_t, b_t) \leq W_t(b_t, s_t)$. Hence, to show that

$$B_t(s_t, b_t) \leq -(\Delta(b_t) \vee \Delta(s_t)) + 3W_t(b_t, s_t)$$

we consider the following scenarios:

**(i)** $b_t \in \mathcal{S}_m^\star$ **and** $s_t \notin \mathcal{S}_m^\star$: In that case,

$$\Delta(b_t) = \rho(b_t) - \rho(m+1); \Delta(s_t) = \rho(m) - \rho(s_t)$$

As event $\mathcal{E}$ holds,

$$B_t(s_t, b_t) = -B_t(b_t, s_t) + 2W_t(b_t, s_t) \leq \Delta(s_t, b_t) + 2W_t(b_t, s_t)$$

As $s_t \notin \mathcal{S}_m^\star$,

$$\rho(s_t) \leq \rho(m+1)$$
$$\Delta(s_t, b_t) \leq \rho(m+1) - \rho(b_t) = -\Delta(b_t)$$

.

But as $b_t \in \mathcal{S}_m^\star$, it also holds that $\rho(b_t) \geq \rho(m)$, and $\Delta(s_t, b_t) \leq \rho(s_t) - \rho(m) = -\Delta(s_t)$. Hence $B_t(s_t, b_t) \leq -(\Delta(b_t) \vee \Delta(s_t)) + 2W_t(b_t, c_t) \leq -(\Delta(b_t) \vee \Delta(s_t)) + 3W_t(b_t, c_t)$.

**(ii)** $b_t \notin \mathcal{S}_m^\star$ **and** $s_t \in \mathcal{S}_m^\star$ :

$$\Delta(s_t) = \rho(s_t) - \rho(m+1); \Delta(b_t) = \rho(m) - \rho(b_t)$$

By Property 1,

$$B_t(s_t, b_t) \leq W_t(b_t, s_t) \leq \hat{\Delta}_t(b_t, s_t) + W_t(b_t, s_t) = B_t(b_t, s_t)$$

as $\hat{\rho}_t(b_t) \geq \hat{\rho}_t(s_t)$. Further, as $\mathcal{E}$ holds,

$$B_t(b_t, s_t) = -B_t(s_t, b_t) + 2W_t(b_t, s_t) \leq \Delta(b_t, s_t) + 2W_t(b_t, s_t)$$

As $b_t \notin \mathcal{S}_m^\star$, $\rho(b_t) \leq \rho(m+1)$ and hence $\Delta(b_t, s_t) \leq \rho(m+1) - \rho(s_t) = -\Delta(s_t)$ As $s_t \in \mathcal{S}_m^\star$, $\rho(s_t) \geq \rho(m)$ and hence $\Delta(b_t, s_t) \leq \rho(b_t) - \rho(m) = -\Delta(b_t)$.
Hence $B_t(s_t, b_t) \leq -(\Delta(b_t) \vee \Delta(s_t)) + 2W_t(b_t, c_t) \leq -(\Delta(b_t) \vee \Delta(s_t)) + 3W_t(b_t, c_t)$.

**(iii)** $b_t \notin \mathcal{S}_m^\star$ **and** $s_t \notin \mathcal{S}_m^\star$: We state that there exists a $b \in \mathcal{S}_m^\star$ that belongs to $N_t$. At any time t,

$$M_t \leftarrow s' \sim_{m'} (U_t \cup N_{t-1})^c$$

$$N_t \leftarrow \text{top}_{m'}(M_t \cup N_{t-1}; \hat{\rho}_{(t-1)})$$

Due to the above sampling approach adopted for $N_t$ which captures the next m' arms with the highest means, we posit that $N_t$ captures at least one arm in $\mathcal{S}_m^\star$. Assuming the event $\mathcal{E}$ holds and $b \in \mathcal{S}_m^\star$,

$$W_t(b_t, s_t) \geq B_t(s_t, b_t) \geq B_t(b, b_t)$$

As by the definition of $s_t$, which is one of the most ambiguous arms with **largest gap to** $b_t$ $B_t(s_t, b_t) \geq B_t(b, b_t)$. Hence, $B_t(s_t, b_t) \geq B_t(b, b_t)$. From this and event $\mathcal{E}$ it follows that

$$B_t(s_t, b_t) \geq B_t(b, b_t) \geq \rho(b) - \rho(b_t) \geq \rho(m) - \rho(b_t)$$

. Hence $W_t(b_t, s_t) \geq B_t(s_t, b_t) \geq \Delta(b_t)$. Using event $\mathcal{E}$,

$$B_t(s_t, b_t) \leq \Delta(s_t, b_t) + 2W_t(b_t, s_t) = (\rho(s_t) - \rho(m)) + (\rho(m) - \rho(b_t)) + 2W_t(b_t, s_t) \quad (4)$$

From 4 and since $B_t(s_t, b_t) \geq \Delta(b_t)$,

$$B_t(s_t, b_t) \leq -\Delta(s_t) + \Delta(b_t) + 2W_t(b_t, s_t) \leq -\Delta(s_t) + 3W_t(b_t, s_t)$$

Also from Property 1 and $W_t(b_t, s_t) \geq \Delta(b_t)$, it holds that

$$B_t(s_t, b_t) \leq W_t(b_t, s_t) = -W_t(b_t, s_t) + 2W_t(b_t, s_t) \leq -\Delta(b_t) + 2W_t(b_t, s_t) \leq -\Delta(b_t) + 3W_t(b_t, s_t)$$

Hence $B_t(s_t, b_t) \leq -(\Delta(b_t) \vee \Delta(s_t)) + 3W_t(b_t, c_t)$.

**(iv)** $b_t \in \mathcal{S}_m^\star$ **and** $s_t \in \mathcal{S}_m^\star$: Then there exists a $s \notin S_m^*$ and $s \in U_t$ In that case,

$$\Delta(b_t) = \rho(b_t) - \rho(m+1); \Delta(s_t) = \rho(s_t) - \rho(m+1)$$

Also by definition of $b_t$ and $s_t$, it holds that $B_t(s_t, b_t) = \max_{i \in U_t} \max_{j \in N_t} [B_t(j, i)]$ Since there exists $s \in U_t$ and $s_t \in N_t$,

$$B_t(s_t, b_t) = \max_{i \in U_t} \max_{j \in N_t} [B_t(j, i)] \geq \max_{j \in N_t} B_t(j, s) \geq B_t(s_t, s) \geq \rho(s_t) - \rho(s) \geq \rho(s_t) - \rho(m+1)$$

As $\rho(s_t) - \rho(m+1) = \Delta(s_t)$, $B_t(s_t, b_t) \geq \Delta(s_t)$ By property 1, $B_t(s_t, b_t) \leq W_t(b_t, s_t)$. Hence,

$$\Delta(s_t) \leq B_t(s_t, b_t) \leq W_t(b_t, s_t)$$

On event $\mathcal{E}$ it follows that $B_t(s_t, b_t) \leq \rho(s_t) - \rho(b_t) + 2W_t(b_t, s_t)$ as $(B(s_t, b_t) \leq W_t(b_t, s_t)$ Then $\rho(s_t) - \rho(b_t)$ can be expressed as $\rho(s_t) - \rho(m+1) + \rho(m+1) - \rho(b_t)$. hence,

$$B_t(s_t, b_t) \leq \rho(s_t) - \rho(m+1) + \rho(m+1) - \rho(b_t + 2W_t(b_t, s_t) \leq \Delta(s_t) - \Delta(b_t) + 2W_t(b_t, s_t)$$

We already know that $B_t(s_t, b_t) \geq \Delta(s_t)$ resulting in,

$$(a) B_t(s_t, b_t) \leq -\Delta(b_t) + 3W_t(b_t, s_t)$$

Now to prove $B_t(s_t, b_t) \leq -\Delta(s_t) + 3W_t(b_t, s_t)$, we rely on property 1,

$$B(s_t, b_t) \leq W_t(b_t, s_t) \leq -W_t(b_t, s_t) + 2W_t(b_t, s_t)$$

As $W_t(b_t, s_t) \geq \Delta(s_t)$, $-W_t(b_t, s_t) \leq -\Delta(s_t)$.Therefore,

$$(b) B(s_t, b_t) \leq W_t(b_t, s_t) \leq -W_t(b_t, s_t) + 2W_t(b_t, s_t) \leq -\Delta(s_t) + W_t(b_t, s_t) \leq -\Delta(s_t) + 3W_t(b_t, s_t)$$

From (a) and (b) $B_t(s_t, b_t) \leq -(\Delta(b_t) \vee \Delta(s_t)) + 3W_t(b_t, c_t)$. $\qquad \square$

## B    PROOF STRUCTURE FOR THEOREM 1

*Proof.* Combining Lemma 4 with stopping rule $B_t(s_t, b_t) \leq \epsilon$ following Lemma 8 in Réda et al. (2021) directly yields

$$N_{a_t}(t) \leq 4\sigma^2 C_{\delta,t}^2 \max\left(\varepsilon, \frac{\varepsilon + \Delta_{a_t}}{3}\right)^{-2}$$

where $N_{a_t}(t)$ is the number of times arms $a$ is sampled.This is equivalent to the sample complexity term $H^\varepsilon(\mu)$ in Theorem 1. Hence, maximum number of samplings on event $\mathcal{E}$ is upper-bound by $\inf_{u \in \mathbb{R}^{*+}}\{u > 1 + H^\varepsilon(\mu)C_{\delta,u}^2\}$, where $H^\varepsilon(\mu) \triangleq 4\sigma^2 \sum_{a \in [K]} \max\left(\varepsilon, \frac{\varepsilon + \Delta_a}{3}\right)^{-2}$.

$\square$

| Dataset | #Train | #Test | Example Question | Description |
|---------|--------|-------|------------------|-------------|
| **GSM8K** Cobbe et al. (2021) | 7473 | 1319 | The red car is 40% cheaper than the blue car. The price of the blue car is $100. How much do both cars cost? | multi-step arithmetic word problems |
| **AquaRat** Ling et al. (2017) | 97467 | 254 | John found that the average of 15 numbers is 40. If 10 is added to each number then the mean of number is? | multi-step arithmetic word problems |
| **TabMWP** Lu et al. (2023b) | 23059 | 7686 | A newspaper researched how many grocery stores there are in each town. What is the mean of the numbers? | Table based numerical reasoning |
| **FinQA** Chen et al. (2022b) | 6251 | 1147 | what is the percentage change in the the gross liability for unrecognized tax benefits during 2008 compare to 2007? | Table and Text based numerical reasoning |
| **StrategyQA** Geva et al. (2021a) | 1800 | 490 | Does the United States Secretary of State answer the phones for the White House? | multi-step reasoning |

Table 3: Overview of the Complex QA datasets used in this study.

## C  DATASETS DESCRIPTION

An overview of the dataset statistics and examples are shown in Table 3.

**FinQA**: Comprises financial questions over financial reports that require numerical reasoning with structured and unstructured evidence. Here, 23.42% of the questions only require the information in the text to answer; 62.43% of the questions only require the information in the table to answer; and 14.15% need both the text and table to answer. Meanwhile, 46.30% of the examples have one sentence or one table row as the fact; 42.63% has two pieces of facts; and 11.07% has more than two pieces of facts. This dataset has 1147 questions in the evaluation set.

**AquaRat**: It comprises 100,000 algebraic word problems in the train set with dev and test set each comprising 254 problems. The problems are provided along with answers and rationales providing the step-by-step solution to the problem. An examples problem is shown in Table 3.

**TabMWP**: It is a tabular-based math word problem-solving dataset with 38,431 questions. TabMWP is rich in diversity, where 74.7% of the questions in TabMWP belong to free-text questions, while 25.3% are multi-choice. We evaluate on the test set with 7686 problems.

**GSM8K**: This dataset consists of linguistically diverse math problems that require multi-step reasoning. The dataset consists of 8.5K problems and we evaluate on the test set of 1319 questions.

**StrategyQA**: To prove the generality of our approach for reasoning tasks, we evaluate on StrategyQA Geva et al. (2021b), a dataset with implicit and commonsense reasoning questions. Since there is no public test set with ground truth answers, we perform stratified sampling done on 2290 full train set to split into 1800 train and 490 test.

**Metrics**: For TabMWP and StrategyQA we employ cover-EM Rosset et al. (2021); Press et al. (2023), a relaxation of Exact Match metric which checks whether the ground truth answer is contained in the generated answer. This helps handle scenarios where LLM generates "24 kilograms" and the ground truth is "24". For other numerical reasoning datasets, we employ Exact match.

## D  RESULTS USING ALTERNATE OPEN SOURCE LLMs

We also report the performance of exemplars from CASE on open-source models like Mistral-7b and LLama2-7b. The results are shown in Table 4. We observe that the absolute performance across baselines and CASE is lower for smaller LLMs like Llama2 and Mistral-7b when compared to gpt-3.5-turbo or gpt-4o-mini. We observe that this is due to the scale of the Language models as Mistral and LlAMA2 models have 7 billion parameters while gpt-3.5-turbo is of much larger scale and the emergent capabilities like ICL, reasoning capabilities are more pronounced in large scale models Wei et al. (2022).

However, we still observe that CASE leads to reasonable performance gains over other static exemplar selection methods across the smaller open-source LLMs. We also observe that CASE is competitive with instance-level/dynamic exemplar selection methods.

Our main experiments are carried out in an exemplar reuse setup where exemplar selection is done using small open source LLMs and transferred to larger LLMs. This is done to reduce the LLM

inference cost during exemplar selection. This setup also leverages the reasoning and emergent capabilities of large scale LLMs. This philosophy is inspired from the work $\mu$P Yang et al. (2022) where the language model hyperparameters are tuned using smaller LM and transferred to a larger LM for the task under consideration.

| Method | GSM8K | AquaRat | TabMWP | FinQA | StrategyQA |
|---|---|---|---|---|---|
| **Mistral-7B** | | | | | |
| **Instance-level** | | | | | |
| KNN Rubin et al. (2022) | 28.00 | 23.16 | 45.3 | 9.06 | 78.27 |
| MMR Ye et al. (2023b) | 28.97 | 18.11 | 47.61 | 10.11 | 79.95 |
| **Task-level** | | | | | |
| Zero-shot-COT Kojima et al. (2023) | 7.42 | 18.89 | 38.96 | 1.74 | 35.37 |
| Manual Few-shot COT | 22.36 | 14.90 | 41.93 | 3.22 | 62.55 |
| LENS Li & Qiu (2023) | 26.08 | 14.17 | 41.82 | 5.14 | 76.12 |
| **Our Approach** | | | | | |
| CASE | **32.6** | 21.2 | **45.55** | **11.24** | 77.75 |
| **Llama2-7B** | | | | | |
| **Instance-level** | | | | | |
| KNN Rubin et al. (2022) | 22.51 | 23.62 | 43.02 | 10.37 | 76.35 |
| MMR Ye et al. (2023b) | 21.60 | 21.65 | 41.66 | 12.20 | 76.32 |
| **Task-level** | | | | | |
| Zero-shot-COT Kojima et al. (2023) | 6.14 | 6.29 | 12.64 | 1.67 | 53.27 |
| Manual Few-shot COT | 19.26 | 20.47 | 23.62 | 2.87 | 64.29 |
| LENS Li & Qiu (2023) | 17.06 | 19.29 | 33.20 | 6.62 | 73.06 |
| **Our Approach** | | | | | |
| CASE | 21.91 | **24.02** | **44.69** | 9.59 | **77.55** |

Table 4: Results across datasets on MISTRAL-7B and LLAMA-2-7B (5-shot exemplars).

# E    ROBUSTNESS OF EXEMPLARS SELECTED BY CASE

We compare the robustness of CASE to other exemplar selection methods. We measure standard deviation of performance across different subsets of the evaluation set through 10-fold cross validation, as shown in Table 5. We observe that in 3 out of 4 datasets, exemplars chosen by CASE has less variance in task performance when compared to other exemplar selection methods. Exemplars selected through instance-level approaches are not optimized for the task but rather on a per-test-example basis. Consequently, this leads to greater variance in final task performance. Hence, CASE helps select exemplars for the task which are more robust than other static methods or instance-level selection methods.

# F    PROMPTS

We also demonstrate the instructions issued to the LLM for different tasks discussed in this work, along with some exemplars selected using CASE. An example of prompt construction for FinQA is shown in Figure 8. We also showcase example prompts for AquaRat (Figure 7), GSM8K (Figure 6), TabMWP (Figure 9) and StrategyQA (Figure 10).

# G    EXEMPLAR QUALITATIVE ANALYSIS

We provide a qualitative analysis of exemplars and compare the exemplars selected using CASE with exemplars selected using LENS Li & Qiu (2023), the recent state-of-the-art approach. The final set of exemplars chosen by LENS vs CASE for the AquaRat dataset is shown in Table 6. We observe that Question 4 and Question 5 in the set of exemplars chosen by LENS are redundant in that they are very similar problems that require similar reasoning steps and are also similar thematically. Both the

| Datasets | GSM8K | AquaRat | TabMWP | FinQA | StrategyQA |
|---|---|---|---|---|---|
| Zero-Shot COT | ±5.18 | ±7.08 | ±1.84 | ±4.50 | ±4.19 |
| Few-Shot COT | ±4.48 | ±12.03 | ±1.66 | ±4.76 | ±5.67 |
| KNN | ±3.76 | ±5.49 | ±1.27 | ±4.17 | ±4.85 |
| MMR | ±4.00 | ±10.53 | ±1.68 | ±6.10 | ±5.70 |
| Graph Cut | ±6.38 | ±8.18 | ±2.03 | ±5.29 | ±7.62 |
| Facility Location | ±4.23 | ±6.71 | ±1.74 | ±4.94 | ±5.93 |
| LENS | ±5.04 | ±6.67 | ±1.59 | ±5.81 | ±3.98 |
| **CASE** | **±3.47** | ±6.86 | **±0.88** | **±3.72** | **±2.91** |

Table 5: Comparison of robustness of CASE to other approaches. We report standard deviation (lower is better) with scores from different splits of the evaluation set.

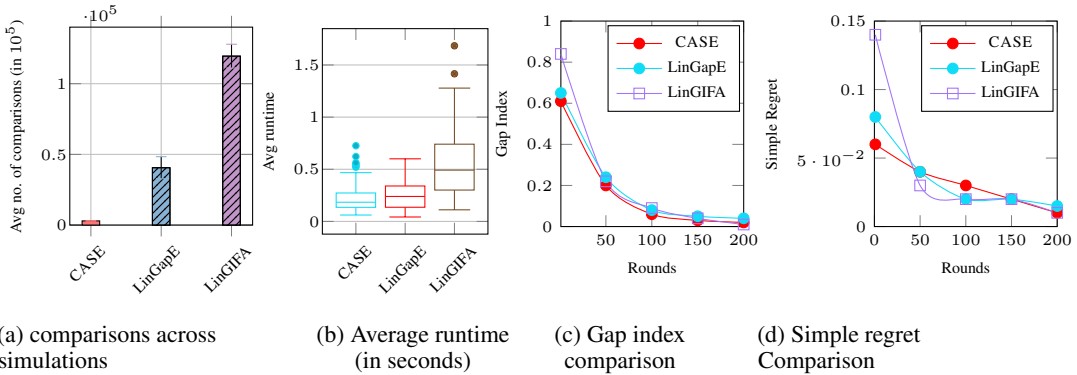

(a) comparisons across simulations

(b) Average runtime (in seconds)

(c) Gap index comparison

(d) Simple regret Comparison

Figure 4: Top-$m$ arm identification by CASE, LinGIFA and LinGapE for K=7, m=3, N=3. (a) Average number of comparisons across simulations (b) Average runtime (in seconds) (c) Gap Index $(B_t(s_t, b_t))$ comparison and (d) Simple regret comparison for each round across simulations

questions are centered on the theme of work and time and are phrased in a similar manner. Hence, they do not add any additional information to solve diverse problems the LLM may encounter during inference. However, we observe that the exemplars chosen by CASE are problems that require diverse reasoning capabilities and are also different thematically.

We also compare the exemplars chosen by CASE with LENS for the FinQA dataset. We observe that the exemplars chosen by CASE comprises diverse set of problems. We also observe that CASE also contains exemplars that require composite numerical operations with multi-step reasoning rationales to arrive at the solutions, whereas LENS mostly has exemplars with single-step solutions.

The exemplars chosen by LENS compared to CASE for TabMWP are shown in Table 9. We observe that exemplar 1 and exemplar 3 chosen by LENS are redundant, as they represent the same reasoning concept of computing median for a list of numbers. However, we observe that CASE selects diverse exemplars, with each exemplar representing a different reasoning concept. We also demonstrate the exemplars for GSM8K and StrategyQA in Table 8 and Table 10 respectively.

# H  SYNTHETIC EXPERIMENTS - EFFICIENCY AND CONVERGENCE ANALYSIS

We further present the results for synthetic experiments for scenarios where $K = 7, m = 3, N = 3$ and $K = 10, m = 3, N = 3$ in Figures 4 and 5. We observe that in both the cases, CASE **drastically reduces** the number of gap-index computations and comparison based operations (comparing arms). For instance, for $K = 10, m = 3$ scenario, on average across 500 simulations CASE only requires 20366.84 comparisons, whereas LinGapE requires 948206.10 comparisons and LinGIFA requires 2180251.73 comparisons. This is due to the shortlist of challenger arms $N_t$ maintained by the proposed approach CASE. We also observe that this results in significant reduction in time (approx. **6x** lower compared to LinGIFA and **2.5x** compared to LinGapE for $K = 10$ case) due to low number of comparisons. We observe that the gap index and simple regret approaches 0 in a similar trend for

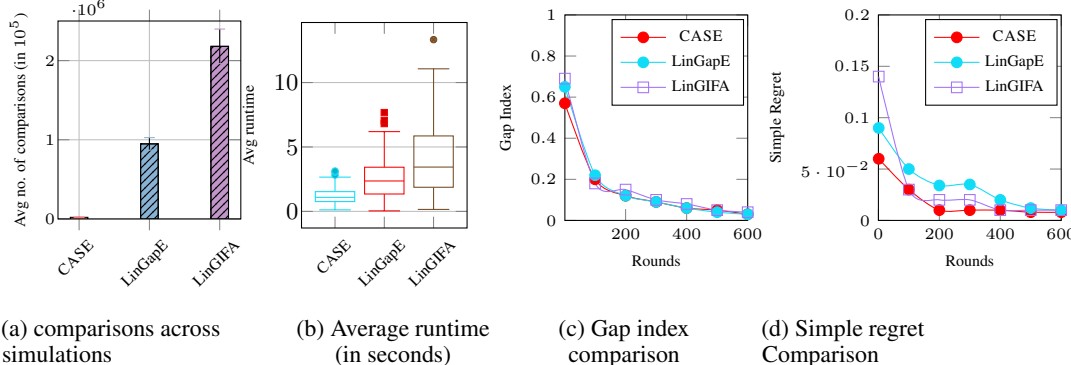

(a) comparisons across simulations

(b) Average runtime (in seconds)

(c) Gap index comparison

(d) Simple regret Comparison

Figure 5: Top-$m$ arm identification by CASE, LinGIFA and LinGapE for K=10, m=3, N=3. (a) Average number of comparisons across simulations (b) Average runtime (in seconds) (c) Gap Index ($B_t(s_t, b_t)$) comparison and (d) Simple regret comparison for each round across simulations

all algorithms. This demonstrates that CASE converges with much lower gap index computations and has lower runtime compared to existing state-of-the-art gap index algorithms.

---

**GSM8K Prompt**

**Instruction**:You are a helpful, respectful and honest assistant helping solve math word tasks requiring reasoning.  Follow the given examples and solve the tasks in step by step manner.

**Exemplars**  :
[Question]: *The red car is 40% cheaper than the blue car. The price of the blue car is $100. How much do both cars cost?*
[Explanation]: The red car is $40/100 * 100 = 40$ cheaper than the blue car.
That means, that the red car costs $100 - 40 = 60$.
So both cars cost $100 + 60 = 160$
[Answer]: 160
...

**Test Input**  : Question:  Explanation: [INS] Answer: [INS]

Figure 6: Prompt for GSM8K

---

**AQUA Prompt**

**Instruction**:You are a helpful, respectful and honest assistant helping solve math word tasks requiring reasoning.  Follow given examples and solve the tasks in step by step manner.

**Exemplars**  :
[Question]: *John found that the average of 15 numbers is 40. If 10 is added to each number then the mean of the number is?*
[Options]: A) 50, B) 45, C) 65, D) 78, E) 64
[Explanation]: $(x0 + x1 + ...x14)/15 = 40$,
new_mean = $40 + 10 = 50$
[Answer]: The option is A
...

**Test Input**  : Question:  Options:  Explanation: [INS] Answer: [INS]

Figure 7: Prompt for AquaRat

---

**FinQA Prompt**

**Instruction**:You are a helpful, respectful and honest assistant helping solve math word tasks requiring reasoning, using the information from given table and text.

**Exemplars** :
*Read the following table, and then answer the question:*
[Table]: beginning balance as of december 1 2007 | 201808
gross increases in unrecognized tax benefits 2013 prior year tax positions | 14009
gross increases in unrecognized tax benefits 2013 current year tax positions | 11350
ending balance as of november 28 2008 | 139549
[Question]: *what is the percentage change in the the gross liability for unrecognized tax benefits during 2008 compare to 2007?*
[Explanation]: $x0 = 139549 - 201808$,
ans = $x0/201808$
[Answer]: -30.9%
. . .

**Test Input:** Read the table and answer the question: Table: Question: Explanation: [INS] Answer: [INS]

Figure 8: Prompt for FinQA

---

**TabMWP Prompt**

**Instruction**:You are a helpful, respectful and honest assistant helping to solve math word problems or tasks requiring reasoning or math, using the information from the given table. Solve the given problem step by step providing an explanation for your answer.

**Exemplars** :
[Table]: Town | Number of stores
Mayfield | 9
Springfield | 9
Riverside | 6
Chesterton | 5
Watertown | 2
[Question]: *A newspaper researched how many grocery stores there are in each town. What is the range of the numbers?*
[Explanation]: Read the numbers from the table. $9, 9, 6, 5, 2$
First, find the greatest number. The greatest number is $9$.
Next, find the least number. The least number is $2$.
Subtract the least number from the greatest number: $9 - 2 = 7$
[Answer]: The range is 7
. . .
. . .

**Test Input** : Table: Question:
Explanation: [INS] Answer: [INS]

Figure 9: Prompt for TabMWP

**StrategyQA Prompt**

**Instruction**:You are a helpful, respectful and honest assistant
helping to solve commonsense problems requiring reasoning.  Follow
the given examples that use the facts to answer a question by
decomposing into sub-questions first and then predicting the final
answer as "Yes" or "No" only.

**Exemplars**  :
[Facts]: The role of United States Secretary of State carries out the President's foreign policy. The
White House has multiple phone lines managed by multiple people.
[Question]: *Does the United States Secretary of State answer the phones for the White House?*
[Sub-question 1]: What are the duties of the US Secretary of State?
[Sub-question 2]: Are answering phones part of #1?

[Answer]: No
. . .
. . .

**Test Input**   : Facts:  Question:
Sub-question: [INS] Answer: [INS]

Figure 10: Prompt for StrategyQA

| Method | Exemplars |
|--------|-----------|
| LENS | **Question**: A cat chases a rat 6 hours after the rat runs. cat takes 4 hours to reach the rat. If the average speed of the cat is 90 kmph, what s the average speed of the rat? **Options**: ['A)32kmph', 'B)26kmph', 'C)35kmph', 'D)36kmph', 'E)32kmph'] **Rationale:** Cat take 10 hours and rat take 4 hours...then Distance is 90*4.so speed of rat is (90*4)/10 = 36kmph **Answer: D** |
| | **Question:** A business executive and his client are charging their dinner tab on the executive's expense account.The ... ? **Options**: ['A)69.55$', 'B)50.63$', 'C)60.95$', 'D)52.15$', 'E)53.15'] **Rationale**: let x is the cost of the food 1.07x is the gross bill after including sales tax 1.15* 1.07x=75 **Answer**: C |
| | **Question:** John and David were each given X dollars in advance for each day they were expected to perform at a community festival. John eventually,... ? **Options**: 'A)11Y', 'B)15Y', 'C)13Y', 'D)10Y', 'E)5Y' **Rationale:** ... **Answer:** A |
| | **Question:** A contractor undertakes to do a piece of work in 40 days. He engages 100 men at the beginning and 100 more after 35 days and completes the work in stipulated time. If he had not engaged the additional men, how many days behind schedule would it be finished?? **Options**: 'A)2', 'B)5', 'C)6', 'D)8', 'E)9' **Rationale:** $[(100\times 35)+(200\times 5)]$ men can finish the work in 1 day therefore 4500 men can finish the work in 1 day. 100 men can finish it in $\frac{4500}{100}$ = 45 days. This is 5 days behind Schedule **Answer:** A |
| | **Question:** A can do a job in 9 days and B can do it in 27 days. A and B working together will finish twice the amount of work in ——- days? **Options**: 'A)22 days', 'B)18 days', 'C)22 6/2 days', 'D)27 days', 'E)9 days' **Rationale:** 1/9 + 1/27= 3/27 = 1/9 9/1 = 9*2 =18 day **Answer:** B |
| CASE | **Question**: In a 1000 m race, A beats B by 50 m and B beats C by 100 m. In the same race, by how many meters does A beat C? **Options**: 'A)156 m', 'B)140 m', 'C)145 m', 'D)169 m', 'E)172 m' **Rationale:** By the time A covers 1000 m, B covers (1000 - 50) = 950 m. By the time B covers 1000 m, C covers (1000 - 100) = 900 m. So, the ratio of speeds of A and C = 1000/950 * 1000/900 = 1000/855. So, by the time A covers 1000 m, C covers 855 m. So in 1000 m race A beats C by 1000 - 855 = 145 m. **Answer: C** |
| | **Question:** Count the numbers between 10 - 99 which yield a remainder of 3 when divided by 9 and also yield a remainder of 2 when divided by 5? **Options**: 'A)Two', 'B)Five', 'C)Six', 'D)Four', 'E)One' **Rationale**: Numbers between 10 - 99 giving remainder 3 when divided by 9 = 12, 21, 30, 39, 48, 57, 66, 75, 84, 93. The Numbers giving remainder 2 when divided by 5 = 12, 57 = 2 **Answer**: A |
| | **Question:** A train running at the speed of 60 km/hr crosses a pole in 3 seconds. Find the length of the train. **Options**: 'A)60', 'B)50', 'C)75', 'D)100', 'E)120' **Rationale:** Speed = 60*(5/18) m/sec = 50/3 m/sec. Length of Train (Distance) = Speed * Time (50/3) * 3 = 50 meter. **Answer:** B |
| | **Question:** If n is an integer greater than 7, which of the following must be divisible by 3? **Options**: 'A)1. n (n+1) (n-4)', 'B)2. n (n+2) (n-1)', 'C)3. n (n+3) (n-5)', 'D)4. n (n+4) (n-2)', 'E)5. n (n+5) (n-6)' **Rationale:** We need to find out the number which is divisible by three, In every 3 consecutive integers, there must contain 1 multiple of 3. So n+4 and n+1 are same if we need to find out the 3's multiple. replace all the numbers which are more than or equal to three ... **Answer:** D |
| | **Question:** A merchant gains or loses, in a bargain, a certain sum. In a second bargain, he gains 280 dollars, and, in a third, loses 20. In the end he finds he has gained 120 dollars, by the three together. How much did he gain or lose bv the first ? **Options**: 'A)80', 'B)-140', 'C)140', 'D)120', 'E)None' **Rationale:** In this sum, as the profit and loss are opposite in their nature, they must be distinguished by contrary signs. If the profit is marked +, the loss must be -. Let x = the sum required. Then according to the statement x + 280 - 20 = 120. And x = -140. **Answer:** B |

Table 6: Qualitative analysis of exemplars for **AquaRat** dataset selected by LENS vs CASE. Rationale is not completely shown for some questions to conserve space. However, in our experiments all exemplars include rationales.

| Method | Exemplars |
|---|---|
| LENS | **Table**: \| increase ( decrease ) \| average yield \| 2.75% ( 2.75 % ) \| volume \| 0.0 to 0.25 \| energy services \| 2013 \| fuel recovery fees \| 0.25 \| recycling processing and commodity sales \| 0.25 to 0.5 \| acquisitions / divestitures net \| 1.0 \| total change \| 4.25 to 4.75% ( 4.75 % ) \|
**Question**: what is the ratio of the acquisitions / divestitures net to the fuel recovery fees as part of the expected 2019 revenue to increase
**Rationale:** ans=( 1.0 / 0.25 ) **Answer:** The answer is 4 |
| | **Table:** ( in millions ) \| 2009 \| 2008 \| 2007 \| sales and transfers of oil and gas produced net of production andadministrative costs \| -4876 ( 4876 ) \| -6863 ( 6863 ) \| -4613 ( 4613 ) \| . . .
**Question**: were total revisions of estimates greater than accretion of discounts?
**Rationale**: . . . **Answer**: The answer is yes |
| | **Table:** \| 2007 \| 2008 \| change \| capital gain distributions received \| 22.1 \| 5.6 \| -16.5 ( 16.5 ) \| other than temporary impairments recognized \| -.3 ( .3 ) \| -91.3 ( 91.3 ) \| -91.0 ( 91.0 ) \| net gains ( losses ) realized onfund dispositions \| 5.5 \| -4.5 ( 4.5 ) \| -10.0 ( 10.0 ) \| net gain ( loss ) . . .
**Question**: what percentage of tangible book value is made up of cash and cash equivalents and mutual fund investment holdings at december 31 , 2009? **Rationale:** ( 1.4 / 2.2 ) **Answer:** The answer is 64% |
| | **Table:** in millions \| 2009 \| 2008 \| 2007 \| sales \| 5680 \| 6810 \| 6530 \| operating profit \| 1091 \| 474 \| 839 \|
**Question**: north american printing papers net sales where what percent of total printing paper sales in 2009? **Rationale:** x0=( 2.8 * 1000 ), ans=( x0 * 5680 ) **Answer:** The answer is 49% |
| | **Table:** in millions \| december 312015 \| december 312014 \| total consumer lending \| 1917 \| 2041 \| total commercial lending \| 434 \| 542 \| total tdrs \| 2351 \| 2583 \| nonperforming \| 1119 \| 1370 \| . . .
**Question**: what was the change in specific reserves in alll between december 31 , 2015 and december 31 , 2014 in billions? **Rationale:** ( .3 - .4 ) **Answer:** The answer is -0.1 |
| CASE | **Table**: in millions \| total \| balance december 31 2006 \| $ 124 \| payments \| -78 ( 78 ) \| balance december 31 2007 \| 46 \| additional provision \| 82 \| payments \| -87 ( 87 ) \| balance december 31 2008 \| 41 \| payments \| -38 ( 38 ) \| balance december 31 2009 \| $ 3 \|
**Question**: in 2006 what was the ratio of the class a shares and promissory notes international paper contributed in the acquisition of borrower entities interest **Rationale:** ans=( 200 / 400 ) **Answer:** 0.5
**Table:** \| 2018 \| 2017 \| 2016 \| allowance for other funds used during construction \| $ 24 \| $ 19 \| $ 15 \| allowance for borrowed funds used during construction \| 13 \| 8 \| 6 \|
**Question**: by how much did allowance for other funds used during construction increase from 2016 to 2018? **Rationale**: x0=( 24 - 15 ),ans=( x0 / 15 ) **Answer**: 60%
**Table:** ( dollars in millions ) \| 2001 ( 1 ) \| 2000 \| 1999 ( 2 ) \| change 00-01 \| adjusted change 00-01 ( 3 ) \| servicing fees \| $ 1624 \| $ 1425 \| $ 1170 \| 14% ( 14 % ) \| 14% ( 14 % ) \| management fees \| 511 \| 581 \| 600 \| -12 ( 12 ) \| -5 ( 5 ) \| foreign exchange trading \| 368 \| 387 \| 306 \| -5 ( 5 ) \| -5 ( 5 ) \| processing fees and other \| 329 \| 272 \| 236 \| 21 \| 21 \| total fee revenue \| 2832\| 2665 \| $ 2312 \| 6 \| 8 \| **Question**: what is the growth rate in total fee revenue in 2001? **Rationale:** x0=( 2832 - 2665 ),ans=( x0 / 2665 ) **Answer:** 6.30%
**Table:** \| increase (decrease) \| average yield \| 2.75% (2.75 %) \| volume \| 0.0 to 0.25 \| energy services \| 2013 \| fuel recovery fees \| 0.25 \| recycling processing and commodity sales \| 0.25 to 0.5 \| acquisitions / divestitures net \| 1.0 \| total change \| 4.25 to 4.75% ( 4.75 % ) \| **Question**: what is ratio of insurance recovery to incremental cost related to our closed bridgeton landfill **Rationale:** ans=(40.0/12.0) **Answer:** 3.33
**Table:** $ in millions \| as of december 2018 \| as of december 2017 \| fair value of retained interests \| $ 3151 \| $ 2071 \| weighted average life ( years ) \| 7.2 \| 6.0 \| constant prepayment rate \| 11.9% ( 11.9 % ) \| 9.4% ( 9.4 % ) \| impact of 10% ( 10 % ) adverse change \| $ -27 ( 27 ) \| $ -19 ( 19 ) \| impact of 20% ( 20 % ) adverse change \| $ -53 ( 53 ) \| $ -35 ( 35 ) \| discount rate \| 4.7% ( 4.7 % ) \| 4.2% ( 4.2 % ) \| impact of 10% ( 10 % ) adverse change \| $ -75 ( 75 ) \| $ -35 ( 35 ) \| impact of 20% ( 20 % ) adverse change \| $ -147 ( 147 ) \| $ -70 ( 70 ) \| **Question**: what was the change in fair value of retained interests in billions as of december 2018 and december 2017? **Rationale:** ans=( 3.28 - 2.13 ) **Answer:** 1.15 |

Table 7: Qualitative analysis of exemplars for **FinQA** dataset selected by LENS vs CASE. Rationale is not completely shown for some questions to conserve space. However, in our experiments all exemplars include rationales.

| Method | Exemplars |
|---|---|
| LENS | **Question**: Michael wants to dig a hole 400 feet less deep than twice the depth of the hole that his father dug. The father dug a hole at a rate of 4 feet per hour. If the father took 400 hours to dig his hole ... ? 
 **Rationale:** Since the father dug a hole with a rate of 4 feet per hour, if the father took 400 hours digging the hole, he dug a hole 4*400 = 1600 feet deep. ... Michael will have to work for 2800/4 = 700 hours. **Answer:** 700 |
| | **Question:** When Erick went to the market to sell his fruits, he realized that the price of lemons had risen by 4 for each lemon. The price of grapes had also increased by half the price that ... ? 
 **Rationale**: The new price for each lemon after increasing by 4 is $8 + 4 = 12$ For the 80 lemons, ... Erick collected $140 * 9 = 1260$ From the sale of all of his fruits, Erick received $1260 + 960 = 2220$. **Answer**: 2220 |
| | **Question:** James decides to build a tin house by collecting 500 tins in a week. On the first day, he collects 50 tins. On the second day, he manages to collect 3 times that number. ... ? 
 **Rationale:** On the second day, he collected 3 times the number of tins he collected on the first day, which is $3 * 50 = 150$ tins. ... he'll need to collect $200/4 = 50$ tins per day to reach his goal. **Answer:** 50 |
| | **Question:** Darrel is an experienced tracker. He can tell about an animal by the footprints it leaves behind. Based on the impressions, he could tell the animal was traveling east at 15 miles/hour ... ? 
 **Rationale:** If we let x be the amount of time, in hours, it will take for Darrel to catch up to the coyote, ... If we subtract 1 x from each side, we get x=1, the amount of time in hours. **Answer:** 1 |
| | **Question:** Martha needs to paint all four walls in her 12 foot by 16 foot kitchen, which has 10 foot high ceilings ... If Martha can paint 40 square feet per hour, how many hours will it take her to paint kitchen? **Rationale:** There are two walls that are 12' by 10' and two walls that are 16' by 10' ... how many hours she needs to finish: 1680 sq ft / 40 sq ft/hour = 42 hours **Answer:** 42 |
| CASE | **Question**: Each class uses 200 sheets of paper per day. The school uses a total of 9000 sheets of paper every week. If there are 5 days of school days, how many classes are there in the school? 
 **Rationale:** Each class uses 200 x 5 = 1000 sheets of paper in a week. Thus, there are 9000/1000 = 9 classes in the school. **Answer:** 9 
 **Question:** If Jenna has twice as much money in her bank account as Phil does, and Phil has one-third the amount of money that Bob has in his account, and Bob has \$60 in his account, how much less money does Jenna have in her account than Bob? **Rationale**: If Phil has one-third of the amount that Bob does, so he has \$60/3= \$20 in his account. Since Jenna has twice as much money as Phil, so she has \$20*2= 40 in her account. Since Bob has \$60 in his account, so he has \$60-\$40=\$20 more than Jenna. **Answer**: 20 
 **Question:** Carlos bought a box of 50 chocolates. 3 of them were caramels and twice as many were nougats. The number of truffles was equal to the number of caramels plus 6. ... If Carlos picks a chocolate at random, what is the percentage chance it will be a peanut cluster? **Rationale:** First find the number of nougats by doubling the number of caramels: 3 caramels * 2 nougats/caramel = 6 nougats. Then find the number of truffles by adding 7 to the number of caramels: 3 caramels + 6 = 9 ... **Answer:** 64 
 **Question:** Janet has 60 less than four times as many siblings as Masud. Carlos has 3/4 times as many siblings as Masud. If Masud has 60 siblings, how many more siblings does Janet have more than Carlos? **Rationale:** If Masud has 60 siblings, and Carlos has 3/4 times as many siblings as Masud, Carlos has 3/4*60=45 siblings. Four times as many siblings as Masud has is 4*60=240. Janet has 60 less than four times as many siblings as Masud, a total of 240-60=180 siblings. ... 180-45=135 **Answer:** 135 
 **Question:** Gavin has had 4 dreams every day for a year now. If he had twice as many dreams last year as he had this year, calculate the total number of dreams he's had in the two years. **Rationale:** If Gavin has been having 4 dreams every day for a year now, he has had 4*365 = 1460 dreams this year. Gavin had twice as many dreams last as he had this year, meaning he had 2*1460 = 2920 dreams last year. The total number of dreams he has had in the two years is 2920+1460=4380 dreams. **Answer:** 4380 |

Table 8: Qualitative analysis of exemplars for **GSM8K** dataset selected by LENS vs CASE. Rationale is not completely shown for some questions to conserve space. However, in our experiments all exemplars include rationales.

| Method | Exemplars |
|---|---|
| LENS | **Table**: \| Name \| Age (years) \| Jessica \| 2 \| Dalton \| 7 \| Kelsey \| 5 \| Lamar \| 8 \| Alexis \| 2 **Question**: A girl compared the ages of her cousins. What is the median of the numbers? **Rationale:** Read the numbers from the table: 2, 7, 5, 8, 2. First, arrange the numbers from least to greatest: 2, 2, 5, 7, 8. Now find the number in the middle. The number in the middle is 5. The median is 5. **Answer:** 5 |
| | **Table:** \| City \| Number of houses sold \| Melville \| 878 \| New Hamburg \| 871 \| Charles Falls \| 881 \| Pennytown \| 817 **Question**: A real estate agent looked into how many houses were sold in different cities. Where were the fewest houses sold? **Rationale**: Find the least number in table. . . . The least number is 817. Now find the corresponding city. Pennytown corresponds to 817. **Answer**: 817 |
| | **Table:** \| Day \| Number of new customers \| Saturday \| 2 \| Sunday \| 2 \| Monday \| 9 \| Tuesday \| 4 \| Wednesday \| 10 \| Thursday \| 3 \| Friday \| 6 **Question**: A cable company analyst paid attention to how many new customers it had each day. What is the median of the numbers? **Rationale:** . . . Find the number in the middle. The number in the middle is 4. The median is 4. **Answer:** 4 |
| | **Table:** \| Day \| Number of cups \| Friday \| 8 \| Saturday \| 4 \| Sunday \| 10 \| Monday \| 6 \| Tuesday \| 6 \| Wednesday \| 1 \| Thursday \| 0 **Question**: Nancy wrote down how many cups of lemonade she sold in the past 7 days. What is the range of the numbers? **Rationale:** Read the numbers from the table: 8, 4, 10, 6, 6, 1, 0. . . . Subtract the least number from the greatest number: 10-0=10. The range is 10. **Answer:** 10 |
| | **Table:** \| Price \| Quantity demanded \| Quantity supplied \|$700 \| 9,800 \| 22,600 \|$740 \| 8,000 \| 22,800 \|$780 \| 6,200 \| 23,000 \|$820 \| 4,400 \| 23,200 \|$860 \| 2,600 \| 23,400 **Question**: At a price of $860, is there a shortage or a surplus? **Rationale:** At price of $860, quantity demanded is less than quantity supplied. . . . So, there is a surplus. **Answer:** surplus |
| CASE | **Table**: Number of siblings \| Frequency 0 \| 19 1 \| 12 2 \| 13 3 \| 9
**Question**: The students in Mr. Robertson's class recorded the no. of siblings that each has. How many students have fewer than 2 siblings? **Rationale:** Find the rows for 0 and 1 sibling. Add the frequencies for these rows. 19 + 12 = 31, 31 students have fewer than 2 siblings. **Answer:** 31
**Table:** \| Apples \| Peaches Organic \| 2 \| 7 Non-organic \| 7 \| 3 **Question**: Brittany conducted a blind taste test on some of her friends in order to determine if organic fruits tasted different than non-organic fruits. Each friend ate one type of fruit. What is the probability that a randomly selected friend preferred organic and tasted peaches? **Rationale**: Let A be the event "the friend preferred organic" and B be the event "the friend tasted peaches" . . . **Answer**: Jul-19
**Table:** dance performance ticket \| $29.00 play ticket \| $32.00 figure skating ticket \| $41.00 ballet ticket \| $37.00 opera ticket \| $76.00 orchestra ticket \| $58.00 **Question**: How much money does Hannah need to buy a ballet ticket and 7 orchestra tickets? **Rationale:** Find the cost of 7 orchestra tickets. $58.00 * 7 = $406.00 . Now find the total cost. $37.00 + $406.00 = $443.00. Hannah needs $443.00. **Answer:** 443
**Table:** Price \| Quantity demanded \| Quantity supplied $665 \| 15,500 \| 16,200 $855 \| 13,700 \| 17,300 $1,045 \| 11,900 \| 18,400 $1,235 \| 10,100 \| 19,500 $1,425 \| 8,300 \| 20,600 **Question**: Look at the table. Then answer the question. At a price of $1,045, is there a shortage or a surplus? **Rationale:** At the price of $1,045, the quantity demanded is less than the quantity supplied. There is too much of the good or service for sale at that price. So, there is a surplus. **Answer:** surplus
**Table:** Comfy Pillows Resort \| 4:15 A.M \| 2:30 P.M \| 10:00 P.M Skyscraper City \| 4:45 A.M \| 3:00 P.M \| 10:30 P.M Pleasant River Campground \| 5:15 A.M \| 3:30 P.M \| 11:00 P.M Rollercoaster Land \| 5:45 A.M \| 4:00 P.M \| 11:30 P.M Floral Gardens \| 6:45 A.M \| 5:00 P.M \| 12:30 A.M Chickenville \| 7:15 A.M \| 5:30 P.M \| 1:00 A.M Happy Cow Farm \| 7:45 A.M \| 6:00 P.M \| 1:30 A.M
**Question**: Look at the following schedule. Marshall got on the train at Rollercoaster Land at 5.45 A.M. What time will he get to Floral Gardens? **Rationale:** Find 5:45 A.M. in the row for Rollercoaster Land. That column shows the schedule for the train that Marshall is on. Look down the column until you find the row for Floral Gardens. Marshall will get to Floral Gardens at 6:45 A.M. **Answer:** 6:45 A.M. |

Table 9: Qualitative analysis of exemplars for **TabMWP** dataset selected by LENS vs CASE. Rationale is not completely shown for some questions to conserve space. However, in our experiments all exemplars include rationales.

| Method | Exemplars |
|--------|-----------|
| LENS | **Facts**: Penguins are native to the deep, very cold parts of the southern hemisphere. Miami is located in the northern hemisphere and has a very warm climate.
**Question:** Would it be common to find a penguin in Miami?
**Rationale:** Where is a typical penguin's natural habitat? What conditions make #1 suitable for penguins? Are all of #2 present in Miami? **Answer: No** |
| | **Facts:** Shirley Bassey recorded the song Diamonds are Forever in 1971. Over time, diamonds degrade and turn into graphite. Graphite is the same chemical composition found in pencils.
**Question:** Is the title of Shirley Bassey's 1971 diamond song a true statement? **Rationale**: What is the title to Shirley Bassey's 1971 diamond song? Do diamonds last for the period in #1? **Answer**: No |
| | **Facts:** The first six numbers in the Fibonacci sequence are 1,1,2,3,5,8. Since 1 is doubled, there are only five different single digit numbers. **Question:** Are there five different single-digit Fibonacci numbers?
**Rationale:** What are the single-digit numbers in the Fibonacci sequence? How many unique numbers are in #1? Does #2 equal 5? **Answer:** Yes |
| | **Facts:** Katy Perry's gospel album sold about 200 copies. Katy Perry's most recent pop albums sold over 800,000 copies. **Question:** Do most fans follow Katy Perry for gospel music? **Rationale:** What type of music is Katy Perry known for? Is Gospel music the same as #1? **Answer:** No |
| | **Facts:** The Italian Renaissance was a period of history from the 13th century to 1600. A theocracy is a type of rule in which religious leaders have power. Friar Girolamo Savonarola was the ruler of Florence, after driving out the Medici family, from November 1494 â€ 23 May 1498.
**Question:** Was Florence a Theocracy during Italian Renaissance? **Rationale:** When was the Italian Renaissance?When did Friar Girolamo Savonarola rule Florence? Is #2 within the span of #1? Did Friar Girolamo Savonarola belong to a religious order during #3? **Answer:** Yes |
| CASE | **Facts**: U2 is an Irish rock band that formed in 1976. The Polo Grounds was a sports stadium that was demolished in 1964. **Question**: Did U2 play a concert at the Polo Grounds? **Rationale:** When was U2 (Irish rock band) formed? When was the Polo Grounds demolished? Is #1 before #2? **Answer:** No |
| | **Facts**: The capacity of Tropicana Field is 36,973. The population of Auburn, NY is 27,687. **Question:** Can you fit every resident of Auburn, New York, in Tropicana Field? **Rationale**: What is the capacity of Tropicana Field? What is the population of Auburn, NY? Is #1 greater than #2? **Answer**: Yes |
| | **Facts**: Door to door advertising involves someone going to several homes in a residential area to make sales and leave informational packets. . . . **Question:** During the pandemic, is door to door advertising considered inconsiderate? **Rationale:** What does door to door advertising involve a person to do? During the COVID-19 pandemic, what does the CDC advise people to do in terms of traveling? . . . Does doing #1 go against #2 and #3? **Answer:** Yes |
| | **Facts:** Mosquitoes cannot survive in the climate of Antarctica. Zika virus is primarily spread through mosquito bites. **Question:** Do you need to worry about Zika virus in Antarctica? **Rationale:** What animal spreads the Zika Virus? What is the climate of Antarctica? Can #1 survive in #2? **Answer:** No |
| | **Facts:** Bob Marley had 9 children. Kublai Khan had 23 children. Many of Bob Marley's children became singers, and followed his themes of peace and love. The children of Kublai Khan followed in his footsteps and were fierce warlords. **Question:** Could Bob Marley's children hypothetically win tug of war against Kublai Khan's children? **Rationale:** How many children did Bob Marley have? How many children did Kublai Khan have? Is #1 greater than #2? **Answer:** No |

Table 10: Qualitative analysis of exemplars for **StrategyQA** dataset selected by LENS vs CASE. Rationale is not completely shown for some questions to conserve space. However, in our experiments all exemplars include rationales.

