# OpenReview forum: "CASE: Challenger Arm Sampling for Efficient In-Context Reasoning"
_ICLR.cc/2025/Conference — Submitted to ICLR 2025_

### Official Review · Reviewer_6HR2 · 2024-10-28

**Soundness:** 3
**Presentation:** 3
**Contribution:** 3
**Rating:** 6
**Confidence:** 3

**Summary:**

Due to the financial and computational costs associated with prcessing large contexts, providing all training examples for in-context learning is often impractical in many settings. This work aims to address this issue by designing an efficient and optimal selection of exemplars to include in the prompt. The authors frame the task of constructing an optimal prompt as a multiple examplar subset selection problem, where, given a training set, the goal is to pick a subset of size m such that the corresponding prompt which is generated maximizes the total validation accuracy. As this is a discrete optimization problem over an exponentially-large search space, naive approaches are intractable.

The authors therefore model this problem as a top-m arm selection problem from the literature on stochastic linear bandits, where they use a linear function to approximate the loss of each arm (i.e. prompt). While the problem of identifying the top-m arms can be solved using gap-index based algorithms, the total number of arms is exponentially large, and so applying these approaches off-the-shelf is computationally infeasible. Instead, the authors propose a new algorithm (CASE) to solve this problem. At a high level, CASE iteratively creates a low-regret set of selected “challenger” arms from uniformly sampled arms. An off-the-shelf algorithm is then used to pick an arm from the union of this set and the current estimate of the top-m arms. The authors prove a high-probabiliyt upper bound for the sample complexity of their algorithm.

The authors apply their algorithm on the GSM8K and AquaRAT datasets for commonsense reasoning, StrategyQA for tabular reasoning, and FinQA & TabMWP for numerical reasoning. They compare their method to zero-shot and manual few-shot baselines, and other instance-level selection measures which use diversity & similarity-based measures to select exemplars for each test example. Across all datasets, the authors find that CASE consistently outperforms random, zero-shot, and few-shot exemplar selection methods. Moreover, CASE (sometimes paired with an existing selection measure) outperforms the other instance-level selection measures on their own. The authors also find that CASE is more computationally-efficient than other stochastic linear bandit-based approaches, and observe that the number of LLM calls made by CASE is significantly less than the number made by LENS (another exemplar selection method). Finally, the authors show that using their exemplar selection methods, smaller methods can be made to perform well compared to larger LLMs.

**Strengths:**

In-context learning is now a ubiquitous task for large language models. Methods for exemplar selection have the potential to produce better results for in-context learning without dealing with the large computational overhead which often comes with feeding the entire training set into the LLM in-context. While the authors are not the first to study the exemplar selection problem, their solution (or some variant of it) outperforms all baselines and other methods across a wide variety of tasks.

**Weaknesses:**

It was unclear to me how the theoretical assumptions in Theorem 1 and Lemma 1 translated to the setting of in-context learning.

While the experimental results are comprehensive, it appears to me that several important baselines are missing. (See Questions for more details.)

**Questions:**

Could you clarify how the assumptions in Theorem 1 and Lemma 1 translate to the setting of in-context learning?

Why didn't you compare to LENS+MMR+SC and LENS+KNN+SC? This is especially relevant given that CASE+MMR+SC and CASE+KNN+SC often performed best across different instances.

---

> ### Author Response · Authors · 2024-11-16
> **Response to Reviewer 6HR2**
>
> Thank you for your insightful comments and highlighting the strengths of our paper. Please find below our responses:
>
>                        TRANSLATION OF THEOREM 1 AND LEMMA 1 TO IN-CONTEXT LEARNING
> > - In the ICL (In-context learning) setting, each arm corresponds to a subset of exemplars. Our goal is to find top-$m$ arms (subsets of exemplars). At each iteration when an arm is pulled or sampled, we call the LLM with exemplars in that arm to get the true mean estimate (**line 22 of Algorithm 1**). We don't have any other particular assumptions for the ICL setting. The $\Delta_a$ (true gap, Theorem 1) depends on the dataset and the LLMs employed.  $K$  is size of $U_T \cup N_T$.
> > - **Theorem 1** helps derive an upper bound on the sample complexity of the multi-arm bandit algorithm employed for exemplar selection in this paper. Here, sample complexity refers to the **expected total number of arms pulled** made by the algorithm before it terminates. In the context of real-world complex reasoning tasks, this directly translates to the **number of LLM calls** required. As stated in Sections 3.1 and 3.2, we model a subset of exemplars as an arm, and pulling an arm involves constructing a prompt using the exemplars within that subset and then calling the LLM to get actual accuracy (reward for that arm) on a validation subset. We have also **updated this information** in the revised manuscript in lines **347-349**.
> > - **Lemma 1** aids in deriving the sample complexity bound for Theorem 1 by bounding the gap-index between $c$ and $b$ (the most confusing arms from $U_t$ and $N_t$), which is tied to the stopping criterion. Our proof of this lemma is based on the approach used by Reda et al., but it is specifically adapted to our CASE algorithm, which maintains a shortlist of challenger arms.
> > - **In summary**, Theorem 1 and Lemma 1 together provide an upper bound on the expected number of arm pulls required by the algorithm, which translates to the number of LLM calls needed when applying CASE to complex reasoning tasks. Thank you for this query, and we will make this discussion clear in the revised version of the manuscript. We hope this response addresses your concern.
>
>
>
>
>
>
>                                 COMPARISON TO LENS+MMR+SC AND LENS+KNN+SC
>
> > - Unlike existing instance-level selection methods, such as KNN and MMR, which select each exemplar independently, without considering the positive and negative interactions between the exemplars. CASE scores exemplar subsets as a whole rather than scoring exemplars independently. Similarly, LENS also selects exemplars independently, assigning a score to each exemplar without accounting for interactions among them.
> > - We would like to highlight that CASE+MMR+SC and CASE+KNN+SC represent a unique **static/dynamic hybrid** approach that preserves the order of exemplars by selecting an entire subset from the top $m$ subsets that has the highest average similarity (for KNN) or maximum marginal relevance (for MMR) to each test example.
> > - Following your suggestion, we extended this static/dynamic hybrid approach to the LENS method by first selecting the top 50 exemplars using LENS and then applying KNN+SC and MMR+SC over this set. The results of these experiments are provided in the table below. We observe from Table that CASE variants with KNN+SC and MMR+SC outperform corresponding LENS (KNN+SC, MMR+SC) variants. We posit that this is because, in CASE, the interaction of exemplars within a selected subset remains preserved. We observe this follows from a similar trend for vanilla CASE, LENS, and CASE + SC, LENS +SC where our proposed CASE variants **outperform** the LENS variants as observed from **Table 1**. We have also **updated these results** in **Table 1** in the uploaded revised manuscript.
> We appreciate your feedback to improve the quality of the work and will include these results in the revised manuscript. We hope this response addresses your concern.
>
>
> | Methods       | GSM8K    | AquaRat  | TabMWP   | FinQA    | StrategyQA |
> |:-------------:|:--------:|:--------:|:--------:|:--------:|:----------:|
> | LENS+KNN+SC   | 80.06    | 57.87    | 79.79    |  60.68   |   83.46    |
> | LENS+MMR+SC   | 80.36    | 59.44  |  80.39     |  60.94   |   82.45    |
> |**CASE+KNN+SC**| **87.49**| **64.17**| **86.23**| **64.25**| **85.92**  |
> |**CASE+MMR+SC**| **85.60**| **62.60**| **85.91**| **63.47**| **84.69**  |
>
> Thank you again for your valuable review. If we have misunderstood any aspect of your review, or if you have additional questions, please let us know.

---

> > ### Comment · Reviewer_6HR2 · 2024-11-21
> >
> > Thanks for your reply, I will keep my score.

---

> ### Author Response · Authors · 2024-11-23
> **Thanks for the response**
>
> Greetings Reviewer 6HR2,
>
>
> Thank you for the response and the valuable review that helped improve the quality of our work. We have also added the above results to the revised manuscript in blue color. We are open to further discussions and are happy to clarify any remaining doubts. If there are still any outstanding concerns, we kindly request you to share them with us.

---

### Official Review · Reviewer_wmwx · 2024-10-29

**Soundness:** 2
**Presentation:** 3
**Contribution:** 2
**Rating:** 5
**Confidence:** 3

**Summary:**

This paper studies how to select a few shot examples (termed exemplars) to be passed with the prompt for the chain-of-thought prompting. They use bandit selection strategies to select the most informative prompts as exemplars. The key issue in this setting is the exponential size of the arm/action set (which is the total set of all available exemplar prompts).  The known bandit strategies for finding the top-m best arm in an exponential action set require the computation of gap indices between all currently estimated top-m arms and the remaining arms. This is impractical in an exponentially large arm set and therefore they propose a new approach to prune the candidate arm set. They basically run a UCB-LCB algorithm over a k-sized subset $S$ which is similar in spirit to algorithms in Chen et al. (2017), and Kaufmann & Kalyanakrishnan (2013). They provide a theoretical guarantee for their algorithm which uses a standard PAC bayesian analysis of fixed confidence top-m bandit theory. They empircally validate their approach on multiple datasets.

**Strengths:**

1. The problem of choosing the best set of informative prompts in-context for reasoning and mathematical analysis for an LLM is an open and interesting problem.
2. Their approach of selecting the top-m best exemplars to be padded with prompt is a theoretically sound approach. However, their practicality for natural language generation needed more validation.
3. They theoretically analyze their algorithms and also empircally validate their approach on multiple datasets.

**Weaknesses:**

1. The writing needs to improve significantly. For example, consider the definition of the UCB, $W_t(i,j)$ definition below eq (3). It is not clear what are $\left(\left\|E\_i\right\|\_{\hat{\Sigma}_t^\lambda}+\left\|E\_j\right\|\_{\hat{\Sigma}_t^\lambda}\right)$. Several such areas require more explanation and definitions need to be defined more clearly.
2. My main concern is that their approach looks mainly like a bandit selection approach without taking into consideration how this affects the natural language generation. For example, their synthetic experiments consider only the bandit case and they show that their approach performs well. However, they then progress to complex reasoning tasks, and it is not clear to me how this bandit approach actually helps the LLM to solve the complex reasoning task. See questions for more.
3. It is not clear to me how the featurization of the actions $\mathcal{X}$ and the $k$-subsets of $\mathcal{X}$ are obtained.
4. How do you learn the $\alpha_i$ in eq(3)? Is it pre-specified?
5. While the $\phi_u(a)$ is estimated using a pre-trained transformer, how do you estimate the featurization $x_i \in \mathcal{X}$?

**Questions:**

1. One of my main concern is that the algorithm does not take into account the reasoning aspect of an LLM while selecting the exemplar. Consider the synthetic experiment vs the reasoning experiment. How do we actually understand the chosen exemplar leads to better reasoning? Consider the example on pg 23 where they show the rationale selected by CASE in the exemplar. Their closest competitor LENS actually chooses rationales that are more concise (and to me look better) than CASE. Also strangely they show different questions and rationales while comparing against their competitor. I suggest showing a table where the same questions show what CASE vs LENS choose as exemplars/rationales.
2. The bigger point I wanted to highlight is that in no part of their algorithm, I found a way that the natural language generation being taken into account to choose the exemplar. It is also not clear how they factor in the original prompt in the arm features (in context) and how this influences the design matrix in line 22 of the algorithm. Do you initialize with the feature of the original prompt?
3. Some more questions are mentioned in the weakness section.
4. Some small things: Chen et al. (2017); Kaufmann & Kalyanakrishnan (2013) are not uniform sampling algorithms for top-m arms. Quite similar to yours, they rely on UCB-LCB mismatch to figure out the top-m arms in stochastic bandits.
5. How do you choose $\epsilon$ in the algorithm? Should it be given as an input to the algorithm?
6. In figure 2d, I observe that LinGapE is almost similar in simple regret to CASE. This goes back to my earlier doubts. How do you actually differentiate how well one bandit algorithm selection is leading to a better reasoning exemplar selection than another bandit algorithm selection.

---

> ### Author Response · Authors · 2024-11-16
> **Response to Reviewer wmwx (1/3)**
>
> Thank you for your insightful comments in the detailed review and for highlighting the strengths of our paper. Please find below our responses for the questions raised:
>
> **Q1: In no part of the algorithm, natural language generation is taken into account to choose the exemplar.**
>
> > We would like to point out that **calling the LLM/accounting for generation from LLM** is taken into account in our algorithm. Specifically, the call to LLM is performed after the selection rule, where an arm (i.e., a set of exemplars) is sampled (**line 22 in Algorithm 1**). Here, **$A(\pi_r(a(t)),\mathcal{V})$** refers to the computation of accuracy on the validation subset, as explained in **lines 200-205**, which involves LLM reasoning to generate predictions on this validation subset. Thus, the LLM's reasoning and generated outputs are explicitly accounted for within our algorithm. We hope this clarifies how the gap-index-based multi-arm bandit algorithm is adopted for selecting exemplars for complex reasoning. We further explain how the featurization of arms is estimated in the complex reasoning setting in response to **Q6** below. Thank you for your query. We have **incorporated this clarification** into **Section 3** (lines **300-305**) in blue-colored text to make it more explicit in the revised manuscript.
>
> **Q2: CASE does not take into account the reasoning aspect of an LLM while selecting the exemplar. In the synthetic experiment vs the reasoning experiment, how do we actually understand the chosen exemplar leads to better reasoning? Better to show a table where for the same questions show what CASE vs LENS choose as exemplars/rationales**
>
> > We think there is a **slight misunderstanding of our setup** and problem formulation, and we would like to clarify the same as follows:
> > - We would first like to clarify that an exemplar is defined as a triplet *(input, rationale, output)*, as defined in **line 42**. The goal is to select few-shot exemplars from the training set to be used as in-context demonstrations for the generative model (LLM) during inference on the test set for complex reasoning tasks. Therefore, the selected exemplars **may vary across different methods**, depending on the algorithm's capability to identify the optimal exemplars. It is important to note that the rationales are already provided within the training examples for all the benchmarks utilized in our work.
> > - Secondly, the LLM reasoning and output generation on the validation subset is taken into account in our algorithm. Specifically, the call to LLM is performed after the selection rule, where an arm (i.e., a set of exemplars) is sampled (**line 22 in Algorithm 1**). Here, **$A(\pi_r(a(t)),\mathcal{V})$** refers to the computation of accuracy on the validation subset, as explained in **lines 200-205**, which involves LLM reasoning to generate predictions on this validation subset. We have **incorporated this clarification** into **Section 3** (lines **300-305**) in blue-colored text to make it more explicit in the revised manuscript.
> > - Additionally, our results indicate that CASE selects high-quality exemplars, as demonstrated by the qualitative analyses in **Tables 6-10**. As explained in **lines 470-475 and 968-1014**, LENS often selects exemplars with redundancy, where multiple examples require similar reasoning steps and share thematic similarities. For instance, in **Table 6** for AquaRat, **exemplars 4 and 5** chosen by LENS are redundant, both centered on the theme of "work and time." Similarly, in **Table 9** for TabMWP, **exemplars 1 and 3** redundantly focus on calculating the median of a list of numbers, lacking diversity that would otherwise enrich the LLM’s inference capability.
> > - Unlike LENS, which assigns scores to exemplars independently without accounting for interactions, CASE captures the interactions between exemplars by scoring subsets as a whole. Furthermore, it learns weights ($\alpha$) for exemplars based on the subset's performance with respect to the LLM loss approximation error. Hence, this results in selecting more diverse and effective exemplars, leading to superior performance.
>
> **Q3: How do you choose $\epsilon$ in the algorithm? Should it be given as an input to the algorithm?**
>
> > The stopping criterion, $\epsilon$, is a hyperparameter provided as input to the algorithm. We have set $\epsilon$ = 0 for synthetic experiments and $\epsilon$ = 0.1 for real-world experiments. These details are mentioned in **Section 4.1**, on **lines 363 and 408**.

---

> ### Author Response · Authors · 2024-11-16
> **Response to Reviewer wmwx (2/3)**
>
> **Q4: LinGapE is similar to CASE from Figure 2(d). How do you differentiate between different bandit algorithms?**
>
> > - Thank you for your question. We would like to point out that the main advantage of CASE is its **efficiency** compared to existing multi-arm bandit frameworks, as explained in **Section 4.2**. This is demonstrated in **Figures 2(a) and 2(b)**, where **CASE shows significantly lower runtime** compared to LinGapE (by approximately **5.6x**) and LinGIFA (by about **12x**). This efficiency gain is due to fewer arm comparisons and reduced gap-index computations.
> > - In our synthetic experiments, we observed that CASE not only provides comparable performance guarantees in terms of error rates but also significantly outperforms existing gap-index frameworks in efficiency. Thus, for real-world complex reasoning tasks, where other gap-index frameworks become computationally expensive, we utilize CASE for exemplar selection. Consequently, we compare our efficient MAB algorithm, CASE, against state-of-the-art task-level exemplar selection methods like LENS.
> > - Additionally, to the best of our knowledge, *we are the first to formulate exemplar selection for in-context learning (ICL) as a top-$m$ arm selection problem*, which sets our approach apart from existing methods.
>
> **Q5. How do you learn the $\alpha_i$ in eq(3)? Is it pre-specified?**
>
> > The parameter $\alpha$ is learned using a least squares estimator. Initially, the $\alpha$’s are initialized from a normal distribution (**line 8 in Algorithm 1**). Subsequently, when an arm is pulled, its true mean (i.e., the validation accuracy in real-world experiments) is obtained. The values of $\alpha$ are then updated using the least squares estimator as specified in **line 24 of Algorithm 1**.
>
> **Q6. While the $\phi_u(a)$ is estimated using a pre-trained transformer, how do you estimate the featurization $x_i \in \mathcal{X}$?**
>
> Thank you for your query. We compute features for each arm as follows:
> > First, we calculate the **similarity** between the exemplars within an arm and the validation samples, as outlined in **line 226**. This results in a tensor of shape *(number of exemplars, number of validation samples)*. We then average across the validation samples to obtain a **feature vector** of dimension *(1, number of exemplars)* for each arm. Consequently, the **feature matrix** has dimensions *(number of arms, number of exemplars)*, as shown in **Equation 3**. We hope this clarifies how the features are constructed. The number of exemplars within an arm determines the dimension of its corresponding feature vector.
>
>
> **Q7. How the featurization of the actions $\mathcal{X}$ and the $k$-subsets of $\mathcal{X}$ are obtained?**
>
> > **Figure 1** and **Section 4.1** illustrates the process of obtaining the $k$-subsets (arms) from $\mathcal{X}$ and subsequently populating the lists $U_t$ and $N_t$. Additionally, in response to your previous question, we have clarified how features are computed for each arm. We hope this addresses your concerns, and we have made these details clearer in the revised manuscript. We appreciate your feedback to further enhance the quality of the paper.
>
> Thank you again for your valuable review. If we have misunderstood any aspect of your review, or if you have additional questions, please let us know.

---

> ### Author Response · Authors · 2024-11-18
> **Response to Reviewer wmwx (3/3)**
>
> **Q8. Consider the definition of the UCB, $W_t(i,j)$ definition below eq (3). It is not clear what are (|Ei|Σ^tλ+|Ej|Σ^tλ). Several such areas require more explanation and definitions need to be defined more clearly.**
>
> > - Thank you for pointing this out. We apologize this was a typographical error and we have fixed this in the revised version of the uploaded manuscript (lines **238-242**). We hope this clarifies the setup.
>
> We have also clarified other concerns raised in the review. Thank you for the insightful questions and suggestions that helped improve the quality of our work. We hope we have sufficiently addressed your queries and request you to share any additional queries or concerns you may have. Thank you for your time and consideration.

---

> ### Author Response · Authors · 2024-11-23
> **Gentle Reminder**
>
> Greetings Reviewer wmwx,
>
> Thank you for your valuable feedback and constructive comments. In our rebuttal, we have provided clarifications regarding the adoption of bandit framework to reasoning setup by taking LLM reasoning into account. We have added explanations in the revised manuscript in blue color to clarify and address all the concerns raised. We are open to further discussions and are happy to clarify any remaining doubts. If there are still any outstanding concerns, we kindly request you to share them with us.

---

> > ### Comment · Reviewer_wmwx · 2024-11-23
> > **Response to the authors**
> >
> > Hi authors,
> >
> > I have gone over your response.
> >
> > 1. I am happy that we agree that this is just a few-shot learning with the bandit selection strategy.
> > 2. I believe that you should downplay the LLM reasoning part significantly. The rationales are already provided when choosing an exemplar. Moroever, given the bandit algorithm and using the covariance matrix, it selects diverse examples and outperforms Lens. Sure, that is believable.  However, the draft significantly overlays the LLM reasoning part which should not be the case.
> > 3. I will take into consideration the other answers to provide a final rating.

---

> ### Author Response · Authors · 2024-11-24
> **Thank you for the response and consideration**
>
> > - Thank you for your valuable feedback. We **acknowledge** that our approach focuses on selecting few-shot exemplars (I/p,o/p, rationales)  in an efficient manner using CASE  to enhance the performance of LLMs on complex reasoning tasks (as indicated in lines **24-25**, Section 1, Section 3.1 and lines **300-306**).
> We would like to humbly clarify that in **prior works on few-shot in-context learning** [1] , exemplar selection [2] , the process of employing rationales within few-shot examples to guide the LLM in generating both a rationale and the corresponding predictions is commonly referred to as **LLM reasoning (as highlighted in [1, 2, 3, 4])**. We adopt this terminology to maintain consistency with the existing literature.
> In our approach, **rationale generation and output prediction** by the LLM, conditioned on the selected few-shot examples, are explicitly considered when pulling an arm to update the parameters (lines **300-306**). Furthermore, during inference, these exemplars are utilized to prompt the LLM to generate **rationales and predictions** for test samples. Our method selects exemplars that showcase diverse reasoning skills through rationales (analysis in **Tables 6-10**) for respective complex tasks.
> > - Our intention was not to overplay LLM reasoning, but to align with established terminology in the field. We are happy to reword any specific paragraphs that the reviewer thinks are overstating the reasoning aspect. We deeply value your feedback and appreciate the opportunity to refine our work further.
>
> [1] Chain-of-Thought Prompting Elicits Reasoning in Large Language Models, Wei et al. NeurIPS 2022.
>
> [2] COMPLEXITY-BASED PROMPTING FOR MULTI-STEP REASONING, Yao Fu et al. ICLR 2023.
>
> [3] Self-Consistency Improves Chain of Thought Reasoning in Language Models, Wang et al. ICLR 2023.
>
> [4] Large Language Models are Zero-Shot Reasoners, Kojima et al. NeurIPS 2022.

---

> ### Author Response · Authors · 2024-11-29
> **Gentle Reminder**
>
> Greetings Reviewer wmwx,
>
> Thank you for your feedback which helped improve the quality of our work. As the reviewer-author discussion period is nearing its conclusion (**December 2nd at 11:59 pm AoE**), we would like to request if we have addressed all your concerns in our previous responses.
>
>
> We have tried to clarify queries regarding the different parameters of CASE and also regarding how CASE is adopted for LLM based rationale and answer generation for different benchmarks. In our approach, *rationale generation and output prediction* by the LLM (LLM generation), conditioned on the selected few-shot examples, are explicitly considered when pulling an arm to update the parameters (**lines 300-306**). Furthermore, during inference, these exemplars are utilized to prompt the LLM to generate rationales and predictions for test samples.
>
>
> Further, we also clarified the use of terminology related to  ``LLM reasoning”,  in the *previous response* and we have also explicitly stated the core motivation and purpose of our work as efficient **task-level exemplar selection** to improve LLM performance (**lines 72-98**, lines **14-15**, lines **24-25**, Section 3.1). We are also happy to reword any instances to make this more explicit in the final version of the manuscript.
>
>
> We hope that we have addressed all the main concerns raised in the initial review. If your concerns have been addressed, *we politely and respectfully request you to consider revising your initial rating*. Thank you so much for your time, consideration, and effort in reviewing our work.
>
>
> Best regards,
>
> The Authors

---

### Official Review · Reviewer_ZiMF · 2024-11-03

**Soundness:** 3
**Presentation:** 2
**Contribution:** 3
**Rating:** 6
**Confidence:** 4

**Summary:**

This paper considers the problem of exemplar selection for In-Context Learning and In-Context Reasoning using LLM’s. The authors consider a setup where they have access to n (labeled) examples and a collection of subsets of k of of these. They want to find the m subsets that lead to the highest m accuracies on a fixed validation set. They postulate a model for the accuracy of ICL as a function of similarities between a validation set and the input. They then apply best-m pure exploration linear bandits algorithms and demonstrate empirical gains across several methods.

**Strengths:**

I thought the paper touched upon an interesting topic in a novel way combining ICL in LLM literature with bandits. The experimental section seemed well done and they demonstrated signficant gains.

**Weaknesses:**

My main concern (and see below) was the couching of the work in the existing bandit literature and theoretical results given. I also had some concerns about the mapping of the ICL problem to bandits.

**Questions:**

A. I thought the mapping of the problem of finding good exemplars to linear bandits to be an interesting one. However, I had some confusions/questions

a) Can you say more about why pi needs m sets of exemplars, rather than  just a set of k examples? Maybe having an example pi in the text would help this could even be in the experiments section.
b) Are S1…Sm allowed to overlap? Why? It feels like this could really alter the approach (see below).
c) How does the objective in (2) turn into the top-m problem? I.e., there seems to be an assumption that finding the top-m individual sets of size k with high accuracy also identifies the collection of best m sets to go into pi?
d) I liked the use of similarity between the training prompts and the validation. In practice, why not just use this directly instead of trying to find a set of m exemplar sets for all prompts? I.e., why go after the average over the validation set? (Overfitting issues aside of course)
e) I think a diagram on page 4 would help

B. I had a few comments about the linear bandit comparisons.
a) The authors are missing some important papers in the pure exploration linear bandit literature, eg, Fiez '19, Jedra'20, Li'23. One key discussion that is missing is that unlike Xu, and Reda, in the m=1 case, all these works provide algorithms that have matching upper and lower bounds. I believe that Degene et al. 2020 (and maybe Li'23?) could be extended to an algorithm that could solve the top-m problem and achieve an optimal lower bound. However both of these algorithms would still be computationally infeasible. It would be good if the authors could comment on this.
b) I didn’t really understand what Theorem 1 says.  What does epsilon-delta PAC means here? How does this sample complexity scale with a function of m, k, n? What is K? This theory section is woefully lacking.
c) As a comment, is the reduction to linear bandits necessary? You are trying to find the top m sets of size k. It feels like you could do this in a more direct manner. In particular, if a1, a2, …., an were an ordering of the individual arms by rewards as given in 3), you just need to return a1, .. a_{mk} appropriately segmented (if there are no overlaps - otherwise it would be a_{k+m}). Could you employ existing works on coarse ranking, eg Katariya '18?

C. Comments on selective exploration
a) I don’t really see the connection to SETC. I went and read this section of the bandit book again, and the comparison is very unclear. Please clarify.
b) As a result, I struggled with the discussion at the start of 3.4. How does the regret of selective exploration factor in?

D. Experiments:
a) The choice of S seemed to be restricted to 100 rather than all subsets of size k. I found this to be a bit confusion (line 3 and 4)

Fiez '19, Sequential experimental design for transductive linear bandits
Jedra '20, Optimal best-arm identification in linear bandits
Wang '21, Fast pure exploration via Frank-Wolfe
Li' 23, Optimal exploration is no harder than Thompson Sampling
Katariya '18 Adaptive Sampling for Coarse Ranking

---

> ### Author Response · Authors · 2024-11-18
> **Response to Reviewer ZiMF (1/3)**
>
> Thank you for your insightful comments and highlighting the strengths of our paper. Please find below our responses:
>
>                                               PART A
>
> **a) Why $\pi$ needs $m$ sets of exemplars, rather than just a set of $k$ examples?**
>
> > - We introduce $\pi$, a dynamic prompt generator, which operates on top of the task-level exemplars selected by CASE. This is equivalent to CASE+KNN and CASE+MMR, as outlined in **lines 191-195**, and corresponds to the CASE static-dynamic hybrid variants shown in **Table 1** and further explained in **Section 4.1, lines 417-421**.
> > - CASE returns $U$, a set of $m$ subsets where each subset consists of $k$ exemplars. The dynamic prompt generator $\pi$ takes these $m$ subsets and applies methods like MMR or KNN to select a subset of exemplars to include in the prompt for each test instance. This approach reduces the search space at the sample level, as we search over these $m$ subsets instead of all training examples, making CASE a **static/dynamic hybrid method**.
>
> **b) Are S1…Sm allowed to overlap? Why?**
>
> > - Yes, the subsets **are allowed to overlap** since we sample with replacement. This design aligns with our motivation to explore different combinations of exemplars, inspired by literature where in-context learning (ICL) samples have been shown to exhibit both **positive and negative interactions** with each other [1]. The key objective is for each subset to serve as a high-quality prompt, covering diverse aspects of the task. Properly capturing interactions among exemplars is critical, as redundant or negatively interacting exemplars can degrade performance. Our qualitative analysis in **Tables 6-10** demonstrates that exemplars in each subset contribute complementary capabilities.
> > - For instance, in **Table 6**, for AquaRat we observe that **exemplars 4** and **5** in the set of selected exemplars by LENS are redundant; they are highly similar problems that require the same reasoning steps and share a common theme of "work and time." Similarly, in **Table 9**, for TabMWP, **exemplars 1** and **3** selected by LENS redundantly focus on computing the median of a list of numbers, failing to introduce new reasoning concepts that would aid the model in solving diverse problems during inference.
> > - In contrast, CASE consistently selects exemplars that are both thematically and conceptually distinct, ensuring a broader coverage of problem types. Thus, it is evident that the CASE algorithm leads to overall performance improvements, as further demonstrated by its comparisons with existing baselines.
>
> [1] Voronov2024, Mind Your Format: Towards Consistent Evaluation of In-Context Learning Improvements.
>
> **c) How does the objective in (2) turn into the top-m problem?...**
>
> > - Thank you for pointing this out. Yes, we assume that the top-$m$ exemplar subsets (prompts), which have the **highest average accuracy** (equivalent to the subsets with the highest individual accuracy), are also those that maximize the objective function in **Equation (2)**.
> > - Please note that the transformation $\pi$ calculates the best subset among the top-$m$ subsets for a given validation example (static-dynamic hybrid approach as described earlier). For the total accuracy of the prompt generated by $\pi$ (Equation 2) to be high, each subset should maximize the accuracy across most validation examples in $\mathcal{V}$. The top-$m$ objective function assumes that the exemplar subset also performs well on other validation examples, so that the average accuracy of each subset over the considered validation subset is the highest. This assumption inherently filters out subsets that are **highly accurate for a few validation examples** but **highly inaccurate for the majority**.
> > - This assumption is critical due to the complexity involved in the search process. The search space is large—first over all possible sets of exemplars to form subsets, and then over all subsets to identify the top-$m$ subsets. This complexity is discussed in **Section 3.2**, specifically in **lines 209-221**.

---

> ### Author Response · Authors · 2024-11-18
> **Response to Reviewer ZiMF on Linear Bandits (2/3)**
>
> Thank you for your insightful reviews. Below we discuss the remaining questions
>
>
>                                             PART A
>
> **d) In practice, why not just use similarity between validation set and exemplars instead of m exemplar sets?**
>
> > - The pairwise similarity between training prompts and validation examples is a baseline for instance-level (or dynamic) prompt selection. However, this approach often fails to capture the interactions between exemplars within a prompt. Apart from its additional computational overhead during inference, we observe that it does not consistently result in stable performance gains (**Table 1**). The fundamental problem tackled in our work is **task-level exemplar selection**.
> > - In contrast, our work offers a principled exploration over the space of exemplar subsets (arms), where the accuracy on a validation set is treated as the true reward for each arm. Through careful exploration, we model the mean estimates for the arm along with confidence information $W_t(i,j)$.
> > - We also perform ablations as shown in **Table 2**, where we fit the model only once on the validation set without exploration (similar to only considering similarity between validation samples and exemplars), and another ablation with exhaustive evaluation. We find that empirically, such methods perform worse, with reasons outlined in **Section 4.5**.
>
>                                             PART B
>
> **a) Discussion of differences to other pure exploration works and relevant citations.**
>
> > - We appreciate the reviewer’s suggestion regarding optimal exploration techniques in linear bandits. According to Li et al. [2], the main problem tackled is to optimally explore the parameter space of a linear bandit by maintaining a distribution $\lambda$ over the sample set $Z$, which could also represent the set of arms $X$.
> > - However, in our specific scenario, the set of arms is large (i.e., all possible subsets of training data points), making it impractical to maintain a distribution over such an extensive set. This problem is common across algorithms in the papers mentioned. Hence, this class of techniques is not directly applicable and is also computationally expensive compared to the proposed gap index-based approach.
> > - Primarily we focus on **designing an efficient approach for gap-index-based MAB algorithms** building upon GIFA framework for the LLM complex reasoning setup. However, we acknowledge that the proof techniques used for analyzing sample complexity could be applicable to the present problem. We will explore the application of these techniques to improve algorithm design and derive tighter sample complexity bounds in future work. Thank you for highlighting these connections. We have cited and discussed the relevant works in **Section 2.2, lines 145-146** (marked in blue in the revised manuscript).
>
> [2] Optimal exploration is no harder than Thompson Sampling, Zhaoqi Li et al. AISTATS 2024.
>
> **b) I didn’t really understand what Theorem 1 says. What does epsilon-delta PAC mean here? How does this sample complexity scale with a function of $m$, $k$, $n$? What is $K$?**
>
> >  We thank the reviewer for pointing out these issues.
> > - **Clarification on Theorem 1:** Theorem 1 adapts the result from Theorem 2 from Reda 2021 et al. [3] to our setting, where we restrict ourselves to arms in $U_T \cup N_T$. It states that the $\epsilon$-optimal top-$m$ arms from $U_T \cup N_T$ are present in $U_T$ with probability $1-\delta$, if $T> \tau_{\delta}$. We will include this clarification in the revised manuscript.
> > - **Definition of $K$:** $K$ refers to the size of $U_T\cup N_T$. This clarification has been added to **lines 338-340** in the updated manuscript.
> > - **Sample Complexity:** We derive a high-probability upper bound on sample complexity, assuming that the average regret of  $U_T \cup N_T$ is less than $\epsilon$.
> Under this assumption, Theorem 1, describes the condition on stopping time $\tau_{\delta}$, which ensures that $U_T$ contains the $\epsilon$-optimal top-$m$ arms. This clarification has been included in the revised manuscript **(lines 319–323)**.
> > - Furthermore, the connection between Lemma 1 and the sample complexity term in **Theorem 1** is elaborated in **Appendix A and B**.
>
> [3] Top-m identification for linear bandits, Clemence Reda et al. AISTATS 2021.

---

> ### Author Response · Authors · 2024-11-19
> **Response to Reviewer ZiMF (3/3)**
>
> We would like to answer rest of the questions as follows
>
>                                                           PART B
>
>
> **c) As a comment, is the reduction to linear bandits necessary? You are trying to find the top $m$ sets of size $k$. It feels like you could do this in a more direct manner. In particular, if $a_1$, $a_2$, …., $a_n$ were an ordering of the individual arms by rewards as given in 3), you just need to return $a_1$, .. $a_{mk}$ appropriately segmented (if there are no overlaps - otherwise it would be $a_{k+m}$). Could you employ existing works on coarse ranking, eg Katariya '18?**
>
> We thank the reviewer for their valuable comments.
> >  -  Here, $k$ refers to the number of exemplars in a subset rather than the number of “arms.” We cannot construct a prompt using a single exemplar, and hence cannot score a single exemplar using the LLM loss. Therefore, we believe that the setting described in coarse ranking is not directly applicable to our problem. Our contribution is primarily an efficient version of gap-index framework applied to ICL setup for LLMs.
>
> > - The linear bandits framework is necessary for defining the confidence $W_{ij}$, which depends on the design matrix $V$.
>
> > - Additionally, the algorithm in Katariya '18 is linear in the number of arms, which is computationally impractical in our setting due to the large number of arms (subsets of exemplars).
>
>
>
>                                                           PART C
>
> **a) I don’t really see the connection to SETC. I went and read this section of the bandit book again, and the comparison is very unclear. Please clarify.**
>
> > - We mention SETC as it provides an algorithm with a sub-linear regret bound for linear bandits with an exponential number of arms. A key similarity between SETC and CASE is that both employ a uniformly random exploration followed by a selection strategy.
> > - In case of SETC, arms are deselected implicitly by freezing certain dimensions. For CASE, arms are selected within the set $U_T \cup N_T$ by implicitly updating the $\alpha$’s.
>
>
>
> Thank you again for your valuable review. If we have misunderstood any aspect of your review, or if you have additional questions, please let us know.

---

> ### Author Response · Authors · 2024-11-25
> **Gentle reminder and thank you for your time and consideration**
>
> Greetings Reviewer ZiMF,
>
> We thank you for your time and insightful review, which helped improve the quality of our work. We have tried to address the concerns raised in your review. Your suggestions have been thoughtfully incorporated into the revision, and we hope the updated manuscript addresses all the concerns you raised. If you have any further queries or require additional clarifications, we kindly request you to share them with us, and we will be happy to address them promptly. If your concerns have been addressed, we respectfully and politely request for a reconsideration of the score. Thank you once again for your time and consideration. We look forward to hearing from you.

---

> > ### Comment · Reviewer_ZiMF · 2024-11-27
> > **Thank you for your response**
> >
> > Hi,
> >
> > Thank you for your response. It answered all my questions adequately.

---

> ### Author Response · Authors · 2024-11-27
> **Thank you for the consideration**
>
> We are pleased to have resolved all your queries and we politely and respectfully request you to consider revising your score. Thank you once again for your time and effort in reviewing our work.

---

> ### Author Response · Authors · 2024-11-29
> **Gentle reminder**
>
> Greetings Reviewer ZiMF,
>
> Thank you for your insightful feedback and your effort in reviewing our work. We are pleased to have resolved all your queries. As the reviewer-author discussion period is nearing its conclusion (**December 2nd at 11:59 pm AoE**), we would like to request if you have any further concerns.
>
> If we have addressed all your concerns, *we politely and respectfully request you to consider revising your initial score*. Thank you once again for your time and effort in reviewing our work.
>
> Best regards,
>
> The Authors

---

### Official Review · Reviewer_vb9k · 2024-11-04

**Soundness:** 3
**Presentation:** 3
**Contribution:** 2
**Rating:** 6
**Confidence:** 4

**Summary:**

This paper proposes a best k-arm identification algorithm to identify the k-examples that needs to be added into the context of a LLM such that it maximizes the downstream performance. They use GSM8K and Aqua datasets to show decent improvements over other state of the art example selection algorithms. They use relatively smaller/older models such as llama2-7b and mistral 7b as well as newer models such as gpt-4o mini.

**Strengths:**

+ The paper provides a reasonable algorithm and the improvements over other SOTA methods are substantially higher that its unlikely to be noise.
+ The results are well supported by real-world experiments on real-world models.
+ There is extensive discussion about reusing the examples from one LLM to another LLM and thus verifying the transfearability of these methods.

**Weaknesses:**

- The paper only compares against other ICL methods. It would be nice to know what happens when compared against full mode finetuning. Especially for models like llama2-70B this is important, since its open model and can potentially be fully FT-ed on the training data.
- There is no sufficient information about contamination of the training data with evals. In particular, where exactly does the improvements come from: is it the algorithm, is it that in the experiments the learning algorithm learns to identify examples in training data that is present in the test set, or something else? There is very little details and discussion about the exact data setup in the paper.

**Questions:**

- Could you please help address both my questions above?

---

> ### Author Response · Authors · 2024-11-14
> **Response to Reviewer vb9k regarding full fine-tuning (1/2)**
>
> Thank you for your insightful comments and highlighting the strengths of our paper. Please find below our responses:
>
>                      EVALUATIONS ARE FOR ICL, WILL BE NICE TO HAVE FULL FINETUNING
>
> > - Thank you for the question and feedback. We would like to humbly point out that our problem setup is **efficient exemplar selection for in-context learning**, not supervised fine-tuning, which falls outside the scope of our work. ICL allows LLMs to perform tasks by conditioning on a context that includes examples or instructions, without the need for additional fine-tuning, making it efficient and adaptable to new tasks. (**lines 34-36**). Notably, recent research has shown that **many-shot ICL** for complex reasoning can perform **on par with, or even outperform**, fine-tuning approaches [1,2,3].
> > - We would also like to humbly highlight that training a 70 billion parameter model is computationally intensive, requiring **four clusters of eight A100 GPUs** [4]. Even with optimizations like low-rank adaptation, fine-tuning still demands significant resources. Additionally, our work which is firmly centered on the ICL setting aims to improve efficiency and task performance through principle task-level exemplar selection.
> > - Our baselines are prior state-of-the-art exemplar selection methods that focus on the ICL setting. To compare our work with respect to fine-tuned approaches, we have included available LLM fine-tuning results from prior works as well as other supervised state-of-the-art results [4,5,6,7,8,9,10,11,12,13] for comparison on relevant benchmarks. We observe that CASE with gpt-3.5-turbo and gpt-4o-mini surpasses fine-tuning-based methods for both smaller and larger open models. Additionally, CASE with Llama2-7b, which does not require LLM fine-tuning, demonstrates competitive performance against versions of Llama2-7b that have been fine-tuned on specific tasks.
>
> |          **Methods**               |       GSM8K        |    Aquarat      |    TabMWP    |   StrategyQA       |
> |:------------------------------------:|:----------------------:|:-----------------:|:-----------------:|:----------------------:|
> | supervised baseline           | 33.0 (FT GPT-3)[8] |   23.90 [9]      | 43.52 (UnifiedQA)[10]| 73.5 [11] |
> | Fine-tuned Llama2-70b       |          56.8 [5]      |          -            |      57.5 [7]    |        75.15 [4]      |
> | Fine-tuned Llama2-7b         |               -            |    18.00 [12]   |       31.1 [7]   |         68.3 [13]     |
> |  **CASE (Llama2-7b)**       |            21.91        |     24.02         |      44.69      |          77.55         |
> |  **CASE (gpt-3.5-turbo)**   |         **79.91**     |    **54.72**     |   **83.42**    |       **84.49**      |
> |  **CASE (gpt-4o-mini)**      |         **91.13**     |    **73.23**     |   **89.73**    |       **95.92**      |
>
>
> **References**
>
> [1] Many-Shot In-Context Learning, Rishabh Agarwal et al. (https://arxiv.org/abs/2404.11018)
>
> [2] Deeper Insights Without Updates: The Power of In-Context Learning Over Fine-Tuning, Qingyu Yin et al. EMNLP 2024.
>
> [3] Is In-Context Learning Sufficient for Instruction Following in LLMs?, Hao Zhao et al. (https://arxiv.org/pdf/2405.19874)
>
> [4] The ART of LLM Refinement: Ask, Refine, and Trust, Kumar Shridhar et al. NAACL 2024.
>
> [5] MuggleMath: Assessing the Impact of Query and Response Augmentation on Math Reasoning, Chengpeng Li et al. ACL 2024.
>
> [6] neuralmagic/Llama-2-7b-gsm8k-pruned_70
>
> [7] Tora: A Tool-Integrated Reasoning Agent For Mathematical Problem Solving, Zhibhin et al. https://arxiv.org/pdf/2309.17452
>
> [8] Large Language Models are Zero-Shot Reasoners, Kojima et al. NeurIPS 2022.
>
> [9] Measuring and Improving BERT’s Mathematical Abilities by Predicting the Order of Reasoning, Piotr Piękos et al. ACL 2021.
>
> [10] Dynamic Prompt Learning Via Policy Gradient For Semi-Structured Mathematical Reasoning, Pan Lu et al. ICLR 2023.
>
> [11] Palm: Scaling language modeling with pathways, Aakanksha Chowdhery et al. JMLR 2023.
>
> [12] SeCoKD: Aligning Large Language Models for In-Context Learning with Fewer Shots, Weixing Wang et al. (https://arxiv.org/pdf/2406.14208)
>
> [13] The Unreasonable Effectiveness of Easy Training Data for Hard Tasks, Peter Hase et al. ACL 2024.

---

> ### Author Response · Authors · 2024-11-14
> **Response to Reviewer vb9k regarding data setup and source of improvement (2/2)**
>
> Below we answer the second question
>
>                         DATA SETUP AND WHERE DO THE IMPROVEMENTS COME FROM?
>
> > - Thank you for the feedback. We have provided comprehensive details regarding the **data setup** in **Appendix C** and **Table 3** and we will highlight this further in Section 4. Our evaluation is based on widely recognized benchmarks for complex reasoning tasks, each of which has undergone a rigorous curation process as detailed in the respective papers. The dataset is divided into training, validation, and test sets, with **no overlap** between these subsets to the best of our knowledge. The authors of these benchmarks have taken careful measures to deduplicate examples, ensuring data integrity. For instance, strategyQA uses disjoint annotators for curating train and test sets and also performs deduplication after formation of the dataset.
> > - For our experiments, we adopt the exemplar selection setup used in prior works like Auto-CoT and Complex-CoT, along with other baselines. Based on your suggestion, **we also reverified** that the **exemplars** selected using CASE are **not** part of the test set. The performance gains we observe stem entirely from the algorithm itself, as evidenced by our thorough **ablation studies** in **Table 2**. These studies demonstrate a significant drop in performance when key components, particularly the exploration step integral to CASE, are removed.
> > - Furthermore, we have provided the **selected exemplars** in our **open-source repository** and have detailed them in **Tables 6-10**, with additional qualitative analyses in the **Results section** and **Appendix G**. We observe that exemplars selected by CASE are thematically diverse and better represent the task compared to other baselines like LENS, which tend to select conceptually similar exemplars, leading to redundancy (**Section 4.3, lines 470-475**).
> > - For instance, in **Table 6**, for AquaRat we observe that **exemplars 4** and **5** in the set of selected exemplars by LENS are redundant; they are highly similar problems that require the same reasoning steps and share a common theme of "work and time." Similarly, in **Table 9**, for TabMWP, **exemplars 1** and **3** selected by LENS redundantly focus on computing the median of a list of numbers, failing to introduce new reasoning concepts that would aid the model in solving diverse problems during inference.
> > - In contrast, CASE consistently selects exemplars that are both thematically and conceptually distinct, ensuring a broader coverage of problem types. Thus, it is evident that the CASE algorithm leads to overall performance improvements, as further demonstrated by its comparisons with existing baselines.
>
> We will highlight the above discussion in the revised manuscript for clarity. Thank you again for your valuable review. We hope we have addressed your concerns. If we have misunderstood any aspect of your review, or if you have additional questions, please let us know.

---

> ### Author Response · Authors · 2024-11-25
> **Gentle Reminder and thank you for your time and consideration**
>
> Greetings Reviewer vb9k,
>
>  We thank you for your time and insightful review, which helped improve the quality of our work. As the reviewer-author discussion period is closing soon (**November 26 at 11:59 pm AoE**), we would like to gently remind you that we are eagerly awaiting your feedback on our response. We have tried to address the concerns raised in your review and clarified the details regarding the **data setup** and other concerns. If you have any further concerns, we kindly request that you share them with us. If your concerns have been addressed, we respectfully and politely request for a reconsideration of the score. Thank you so much for your time and consideration.

---

> ### Author Response · Authors · 2024-11-29
> **Eagerly waiting for your feedback**
>
> Greetings Reviewer vb9k,
>
> Thank you for your time and insightful feedback. As the reviewer-author discussion period is nearing its conclusion (**December 2nd at 11:59 pm AoE**), we would like to request if we have addressed all your concerns.
>
> We greatly value your insightful comments, which have been instrumental in improving our work. Your suggestions have been thoughtfully incorporated into the revision, and we hope that the author's response and the updated manuscript address all the concerns you raised.
>
> If you have any further queries or require additional clarifications, we kindly request you to share them with us, and we will be happy to address them promptly. If your concerns have been addressed, *we respectfully request you to consider revising your initial score*. Thank you once again for your time and effort in reviewing our work.
>
>
> Best regards,
>
> The Authors

---

> > ### Comment · Reviewer_vb9k · 2024-12-02
> > **I have read the responses**
> >
> > Dear authors,
> >
> > thanks for the detailed results. yes i have already read the responses and taken those into account. Thanks for the followup experiments.

---

### Author Response · Authors · 2024-12-04
**Summary of review discussions and revised manuscript**

*We thank the reviewers and the Area Chairs for their valuable time and insightful feedback, which have significantly improved the quality of our work. We would like to summarize the response, discussion, and revisions made for all the reviews. We have also uploaded the revised version of the manuscript to Openreview with changes highlighted in **blue color.***

> - In response to **Reviewer vb9k**, we clarified the **data setup** pointing to respective sections in our paper. We also clarified that our problem setup primarily focuses on **Efficient In-Context Learning**, noting that **fine-tuning LLMs** is computationally expensive and beyond the scope of the current work. Additionally, we provided a comparison with fine-tuned results from prior works in the author response, showing that CASE achieves superior performance in ICL-based reasoning. This aligns with observations from prior works indicating that, for certain tasks, ICL can match or outperform fine-tuning (references and details provided in our response). The reviewer acknowledged our response and we are glad it helped address the questions raised. The response from the reviewer: **``thanks for the detailed results. yes i have already read the responses and taken those into account. Thanks for the followup experiments."**.

> - In response to **Reviewer ZiMF**, we clarified the linear bandits framework and associated notations, which are also reflected in the revised manuscript in **Section 3.4**. We also incorporated discussions of the relevant works mentioned by the reviewer in **Section 2.2** of the revised manuscript, emphasizing how our gap-index-based approach (CASE) fundamentally differs from these methods. We also clarified other questions on problem formulation and setup in author response.  We are also glad that the reviewer found our response to **``adequately address the concerns"** as indicated in the response from the reviewer: **``Thank you for your response. It answered all my questions adequately**".



> - In response to **Reviewer wmwx**, we clarified a slight misunderstanding regarding the definition of an exemplar and the role of LLM reasoning in Algorithm 1. We explicitly **clarified how LLM generation** was already taken into account in CASE during exemplar selection, specifically at **Line 22 in Algorithm 1**, where when an arm is pulled, the LLM is called with exemplars in that arm to generate rationale with predictions on validation set. This yields accuracy on the validation set and is used as a reward for the arm to update the parameters. We highlighted this in both the author response and the revised manuscript (**Lines 300-306**). We also clarified queries regarding notation in Section 3 and clarified question regarding featurization for linear bandits. We also clarified how $\alpha_i$’s are learned. The reviewer acknowledged our initial response and raised questions regarding the LLM reasoning aspect. In response, we clarified the terminology related to "LLM reasoning" by referencing prior literature [1,2,3]. Specifically, we highlighted that the use of few-shot examples with rationales to prompt the LLM to generate both rationales and predictions is commonly referred to as **LLM reasoning**. We have also explicitly stated the core motivation and purpose of our work as **efficient task-level exemplar selection to improve LLM performance (lines 72-98, lines 14-15, lines 24-25, Section 3.1)**. We use ``LLM reasoning"  terminology to align with existing terminology [1,2,3] in the field and our intention is not to overstate the same which is also evident in our motivation, problem formulation, and experiments. We are also happy to reword any instances to make this further explicit, in the final version of the manuscript.
    - *We thank the reviewer for their valuable feedback to improve the quality of the work. We are also grateful to the reviewer for their acknowledgment of our response to other questions in the initial review and mention of  ``**consideration of these answers to provide the final rating**"*.

> -  In response to **Reviewer 6HR2**, we added the baselines LENS+MMR+SC and LENS+KNN+SC into **Table 1** of the revised manuscript. We also clarified the connection of Theorem 1 to the In-Context Reasoning setup and clarified related assumptions in the author response below. **The reviewer acknowledged our response and there were no further concerns or questions raised.**

*We hope we have answered all the concerns raised by the reviewers and if you have additional questions, please let us know, and we would be happy to engage in further discussion. Thank you once again for your valuable time and constructive feedback!*

[1] Chain-of-Thought Prompting Elicits Reasoning in Large Language Models, Wei et al. NeurIPS 2022.

[2] COMPLEXITY-BASED PROMPTING FOR MULTI-STEP REASONING, Yao Fu et al. ICLR 2023.

[3] Self-Consistency Improves Chain of Thought Reasoning in Language Models, Wang et al. ICLR 2023.

---

### Meta-Review · Area_Chair_NBor · 2024-12-18

**Metareview:**

This paper uses top-$m$ best-arm identification (BAI) for selecting examples for in-context learning (ICL). The proposed solution is an existing bandit algorithm with short-listed candidate examples, to control computational complexity. The algorithm is analyzed and also empirically evaluated. The empirical evaluation is the strongest part of the paper. The scores of this paper are 3x 6 and 5, and did not change during the discussion. The reviewers and I have several concerns:

* **Experiments:** It would be nice to have non-ICL baselines, such as fine-tuning. Fine-tuning will likely outperform ICL when given enough data and that is fine. More complex tasks would also strengthen empirical evaluation.

* **Clarity:** I was surprised that the reviewers did not raise their scores during the rebuttal. Therefore, I read the paper and see why. The writing needs to improve significantly. As an example, $\hat{\rho}_t(a)$ in Algorithm 1 is undefined. $\rho$ has one parameter before (3) and two parameters in (3). This lack of definitions at some places and overload of the notation at others make it hard to understand what is done. The approach needs to be better motivated as well. To start with, you build on GIFA, which is a fixed-confidence BAI algorithm. Why is your setting BAI and not just learning to rank in-context examples? The latter does not require exploration or simple regret.

The lack of clarity in writing is a major concern and therefore the paper cannot be accepted at this time.

**Additional Comments On Reviewer Discussion:**

See the meta-review for details.

---

### Decision · Program_Chairs · 2025-01-22

Reject